# Tractable Function-Space Variational Inference in Bayesian Neural Networks

**Tim G. J. Rudner**[*]
University of Oxford

**Zonghao Chen**
University College London

**Yee Whye Teh**
University of Oxford

**Yarin Gal**
University of Oxford

## Abstract

Reliable predictive uncertainty estimation plays an important role in enabling the deployment of neural networks to safety-critical settings. A popular approach for estimating the predictive uncertainty of neural networks is to define a prior distribution over the network parameters, infer an approximate posterior distribution, and use it to make stochastic predictions. However, explicit inference over neural network parameters makes it difficult to incorporate meaningful prior information about the data-generating process into the model. In this paper, we pursue an alternative approach. Recognizing that the primary object of interest in most settings is the distribution over functions induced by the posterior distribution over neural network parameters, we frame Bayesian inference in neural networks explicitly as inferring a posterior distribution over functions and propose a scalable function-space variational inference method that allows incorporating prior information and results in reliable predictive uncertainty estimates. We show that the proposed method leads to state-of-the-art uncertainty estimation and predictive performance on a range of prediction tasks and demonstrate that it performs well on a challenging safety-critical medical diagnosis task in which reliable uncertainty estimation is essential.

## 1 Introduction

Machine learning models succeed at an increasingly wide range of narrowly defined tasks [Krizhevsky et al., 2012, Mnih et al., 2013, Silver et al., 2016, Jumper et al., 2021] but may fail without warning when used on inputs that are meaningfully different from the data they were trained on [Amodei et al., 2016, Hendrycks et al., 2021, Rudner and Toner, 2021a,b]. To deploy machine learning models in safety-critical environments where failures are costly or may endanger human lives, machine learning methods must be reliable and have the ability to 'fail gracefully.' A promising tool for incorporating fail-safe mechanisms into machine learning systems, predictive uncertainty quantification allows machine learning models to express their confidence in the correctness of their predictions.

In this paper, we develop a method for obtaining reliable uncertainty estimates in Bayesian neural networks (BNNs, Neal [1996]). While BNNs have promised to combine the advantages of deep learning and Bayesian inference, existing approaches for approximate inference in BNNs fall short of this promise and have been demonstrated to result in approximate posterior predictive distributions that underperform 'non-Bayesian' methods both in terms of predictive accuracy and uncertainty quantification—making them of limited use in practice [Ovadia et al., 2019, Foong et al., 2019, Farquhar et al., 2020a, Band et al., 2021]. A potential reason for this shortcoming is that commonly used parameter-space inference methods make it difficult to define meaningful priors that effectively incorporate information about the data-generating process into inference.

---

[*]Corresponding author. Email: `<tim.rudner@cs.ox.ac.uk>`.

36th Conference on Neural Information Processing Systems (NeurIPS 2022).

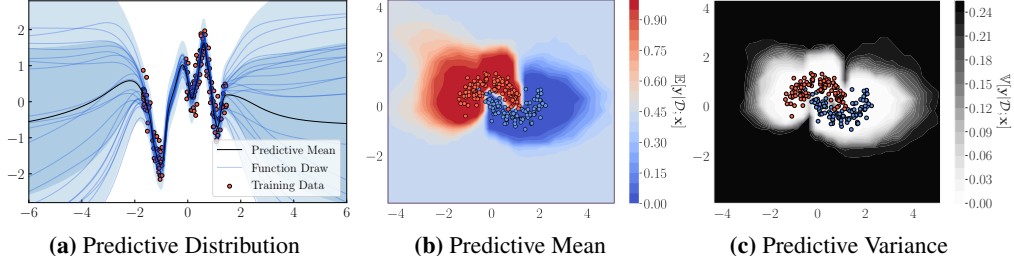

| **(a)** Predictive Distribution | **(b)** Predictive Mean | **(c)** Predictive Variance |

**Figure 1:** 1D regression on the *Snelson* dataset and binary classification on the *Two Moons* dataset. The plots show the predictive distributions of a BNNs, obtained via function-space variational inference (FSVI). For further illustrative exampled and comparisons to deep ensembles and BNNs learned via parameter-space variational inference, see Appendix B.

To avoid this limitation, we follow Sun et al. [2019] and consider a variational objective defined explicitly in terms of distributions over *functions* induced by distributions over parameters. In contrast to prior works that rely on approximation techniques that prevent such function-space variational objectives to be used with high-dimensional inputs and highly-overparameterized neural networks, we propose a simple estimator of the Kullback-Leibler divergence between distributions over functions that enables us to perform stochastic variational inference. The proposed estimation procedure allows defining priors that explicitly encourage high predictive uncertainty away from the training data as well as priors that reflect relevant information about the task at hand.

We demonstrate that this approach leads to posterior approximations that exhibit significantly improved predictive uncertainty estimates compared to a wide array of state-of-the-art Bayesian and non-Bayesian methods. Figure 1 shows examples of predictive distributions obtained via function-space variational inference on low-dimensional, easy-to-visualize datasets. As can be seen in the figures, the predictive distributions fit the training data well while also exhibiting a high degree of predictive uncertainty in parts of the input space far away from the training data, as desired.

**Contributions.** We propose a simple estimation procedure for performing function-space variational inference in BNNs. The variational method allows for the incorporation of meaningful prior information about the data-generating process into the inference and produces reliable predictive uncertainty estimates. We perform a thorough empirical evaluation in which we compare the proposed approach to a wide array of competing methods and show that it consistently results in high predictive performance and reliable predictive uncertainty estimates, outperforming other methods in terms of predictive accuracy, robustness to distribution shifts, and uncertainty-based detection of distributionally-shifted data samples. We evaluate the proposed method on standard benchmarking datasets as well as on a safety-critical medical diagnosis task in which reliable uncertainty estimation is essential.[2]

## 2   Preliminaries

We consider supervised learning tasks on data $\mathcal{D} \doteq \{(\mathbf{x}_n, \mathbf{y}_n)\}_{n=1}^N = (\mathbf{X}_\mathcal{D}, \mathbf{y}_\mathcal{D})$ with inputs $\mathbf{x}_n \in \mathcal{X} \subseteq \mathbb{R}^D$ and targets $\mathbf{y}_n \in \mathcal{Y}$, where $\mathcal{Y} \subseteq \mathbb{R}^Q$ for regression and $\mathcal{Y} \subseteq \{0, 1\}^Q$ for classification tasks. Bayesian neural networks (BNNs) are stochastic neural networks trained using (approximate) Bayesian inference. Denoting the parameters of such a stochastic neural network by the multivariate random variable $\mathbf{\Theta} \in \mathbb{R}^P$ and letting the function mapping defined by a neural network architecture be given by $f : \mathcal{X} \times \mathbb{R}^P \to \mathbb{R}^Q$, we obtain a random function $f(\cdot\,;\mathbf{\Theta})$. For a parameter realization $\boldsymbol{\theta}$, we obtain a corresponding function realization, $f(\cdot\,;\boldsymbol{\theta})$. When evaluated at a finite collection of points $\mathbf{X} = \{\mathbf{x}_i\}_{i=1}^m$, $f(\mathbf{X};\mathbf{\Theta})$ is a multivariate random variable and $f(\mathbf{X};\boldsymbol{\theta})$ is a vector.

Letting $p_{\mathbf{y}|f(\mathbf{X};\mathbf{\Theta})}$ be a likelihood function and $p_{\mathbf{y}|f(\mathbf{X};\mathbf{\Theta})}(\mathbf{y}_\mathcal{D} \,|\, f(\mathbf{X}_\mathcal{D};\boldsymbol{\theta}))$ be the likelihood of observing the targets $\mathbf{y}_\mathcal{D}$ under the stochastic function $f(\cdot\,;\mathbf{\Theta})$ evaluated at inputs $\mathbf{X}_\mathcal{D}$ and letting $p_\mathbf{\Theta}$ be a prior distribution over the stochastic network parameters $\mathbf{\Theta}$, we can use Bayes' Theorem to find the posterior distribution, $p_{\mathbf{\Theta}|\mathcal{D}}$ [MacKay, 1992, Neal, 1996]. However, since the mapping $f$

---

[2]Our code can be accessed at `https://github.com/timrudner/FSVI`.

is a nonlinear function of the stochastic parameters $\boldsymbol{\Theta}$, exact inference is analytically intractable. Variational inference is an approach that seeks to sidestep this intractability by framing posterior inference as a variational optimization problem, where the goal is to find a distribution $q_{\boldsymbol{\Theta}}$ in a variational family $\mathcal{Q}_{q_{\boldsymbol{\Theta}}}$ that solves the variational problem $\min_{q_{\boldsymbol{\Theta}} \in \mathcal{Q}_{q_{\boldsymbol{\Theta}}}} \mathbb{D}_{\mathrm{KL}}(q_{\boldsymbol{\Theta}} \,\|\, p_{\boldsymbol{\Theta}|\mathcal{D}})$ [Wainwright and Jordan, 2008]. If $\mathcal{Q}_{q_{\boldsymbol{\Theta}}}$ is the family of mean-field Gaussian distributions and the prior distribution over parameters $p_{\boldsymbol{\Theta}}$ given by a diagonal Gaussian distribution, the resulting variational objective is amenable to stochastic variational inference and can be optimized using gradient-based methods [Hinton and van Camp, 1993, Graves, 2011, Hoffman et al., 2013, Blundell et al., 2015].

## 2.1 A Function-Space Perspective on Variational Inference in Bayesian Neural Networks

Instead of seeking to infer an approximate posterior distribution over parameters, we frame variational inference in stochastic neural networks as inferring an approximation to the posterior distribution over *functions* $p_{f(\cdot\,;\boldsymbol{\Theta})|\mathcal{D}}$ induced by the posterior distribution over parameters $p_{\boldsymbol{\Theta}|\mathcal{D}}$, that is,

$$p_{f(\cdot\,;\boldsymbol{\Theta})|\mathcal{D}}(f(\cdot\,;\boldsymbol{\theta}) \,|\, \mathcal{D}) = \int_{\mathbb{R}^P} p_{\boldsymbol{\Theta}|\mathcal{D}}(\boldsymbol{\theta}' \,|\, \mathcal{D})\, \delta(f(\cdot\,;\boldsymbol{\theta}) - f(\cdot\,;\boldsymbol{\theta}'))\, \mathrm{d}\boldsymbol{\theta}', \tag{1}$$

where $\delta(\cdot)$ is the Dirac delta function [Wolpert, 1993]. Considering the prior distribution over functions $p_{f(\cdot\,;\boldsymbol{\Theta})}$ induced by a prior distribution over parameters $p_{\boldsymbol{\Theta}}$,

$$p_{f(\cdot\,;\boldsymbol{\Theta})}(f(\cdot\,;\boldsymbol{\theta})) = \int_{\mathbb{R}^P} p_{\boldsymbol{\Theta}}(\boldsymbol{\theta}')\, \delta(f(\cdot\,;\boldsymbol{\theta}) - f(\cdot\,;\boldsymbol{\theta}'))\, \mathrm{d}\boldsymbol{\theta}', \tag{2}$$

and the variational distribution over functions $q_{f(\cdot\,;\boldsymbol{\Theta})}$ induced by a variational distribution over parameters $q_{\boldsymbol{\Theta}}$,

$$q_{f(\cdot\,;\boldsymbol{\Theta})}(f(\cdot\,;\boldsymbol{\theta})) = \int_{\mathbb{R}^P} q_{\boldsymbol{\Theta}}(\boldsymbol{\theta}')\, \delta(f(\cdot\,;\boldsymbol{\theta}) - f(\cdot\,;\boldsymbol{\theta}'))\, \mathrm{d}\boldsymbol{\theta}', \tag{3}$$

we can express the problem of finding a posterior distribution over functions variationally as

$$\min_{q_{\boldsymbol{\Theta}} \in \mathcal{Q}_{q_{\boldsymbol{\Theta}}}} \mathbb{D}_{\mathrm{KL}}(q_{f(\cdot\,;\boldsymbol{\Theta})} \,\|\, p_{f(\cdot\,;\boldsymbol{\Theta})|\mathcal{D}}), \tag{4}$$

which allows us to effectively incorporate meaningful prior information about the underlying data-generating process into training. As discussed by Burt et al. [2021], this variational objective is guaranteed to be well-defined for suitably chosen prior distributions over functions. Specifically, the KL divergence between two distributions over functions generated from different distributions over parameters applied to the same mapping (e.g., the same neural network architecture) is well-defined (i.e., finite) if the KL divergence between the distributions over parameters is finite, since, by the strong data processing inequality [Polyanskiy and Wu, 2017],

$$\mathbb{D}_{\mathrm{KL}}(q_{f(\cdot\,;\boldsymbol{\Theta})} \,\|\, p_{f(\cdot\,;\boldsymbol{\Theta})}) \leq \mathbb{D}_{\mathrm{KL}}(q_{\boldsymbol{\Theta}} \,\|\, p_{\boldsymbol{\Theta}}). \tag{5}$$

As a result, if $\mathbb{D}_{\mathrm{KL}}(q_{\boldsymbol{\Theta}} \,\|\, p_{\boldsymbol{\Theta}}) < \infty$, which is the case for finite-dimensional parameter vectors $\boldsymbol{\Theta}$ and $q_{\boldsymbol{\Theta}}$ absolutely continuous with respect to $p_{\boldsymbol{\Theta}}$, then the function-space KL divergence is finite and thus well-defined as a variational objective.

Hence, for a likelihood function defined on a finite set of training targets $\mathbf{y}_{\mathcal{D}}$ and a suitably defined prior distribution over functions, we can express the variational problem above equivalently as the well-defined maximization problem $\max_{q_{\boldsymbol{\Theta}} \in \mathcal{Q}_{\boldsymbol{\theta}}} \mathcal{F}(q_{\boldsymbol{\Theta}})$ with

$$\mathcal{F}(q_{\boldsymbol{\Theta}}) \doteq \mathbb{E}_{q_{f(\mathbf{X}_{\mathcal{D}};\boldsymbol{\Theta})}}[\log p_{\mathbf{y}|f(\mathbf{X};\boldsymbol{\Theta})}(\mathbf{y}_{\mathcal{D}} \,|\, f(\mathbf{X}_{\mathcal{D}};\boldsymbol{\theta}))] - \mathbb{D}_{\mathrm{KL}}(q_{f(\cdot\,;\boldsymbol{\Theta})} \,\|\, p_{f(\cdot\,;\boldsymbol{\Theta})}), \tag{6}$$

where $\mathbb{D}_{\mathrm{KL}}(q_{f(\cdot\,;\boldsymbol{\Theta})} \,\|\, p_{f(\cdot\,;\boldsymbol{\Theta})})$ is also a KL divergence between distributions over functions.

Unfortunately, evaluating the KL divergence in Equation (6) is in general intractable for arbitrary mappings $f$. To obtain a tractable objective, Sun et al. [2019] showed that $\mathbb{D}_{\mathrm{KL}}(q_{f(\cdot\,;\boldsymbol{\Theta})} \,\|\, p_{f(\cdot\,;\boldsymbol{\Theta})})$ can be expressed as the supremum of the KL divergence from $q_{f(\cdot\,;\boldsymbol{\Theta})}$ to $p_{f(\cdot\,;\boldsymbol{\Theta})}$ over all *finite* sets of evaluation points, resulting in the objective function

$$\mathcal{F}(q_{\boldsymbol{\Theta}}) = \mathbb{E}_{q_{f(\mathbf{X}_{\mathcal{D}};\boldsymbol{\Theta})}}[\log p_{\mathbf{y}|f(\mathbf{X};\boldsymbol{\Theta})}(\mathbf{y}_{\mathcal{D}} \,|\, f(\mathbf{X}_{\mathcal{D}};\boldsymbol{\theta}))] - \sup_{\mathbf{X} \in \mathcal{X}_{\mathbb{N}}} \mathbb{D}_{\mathrm{KL}}(q_{f(\mathbf{X};\boldsymbol{\Theta})} \,\|\, p_{f(\mathbf{X};\boldsymbol{\Theta})}), \tag{7}$$

where $\mathcal{X}_{\mathbb{N}} \doteq \bigcup_{n \in \mathbb{N}}\{\mathbf{X} \in \mathcal{X}_n \,|\, \mathcal{X}_n \subseteq \mathbb{R}^{n \times D}\}$ is the collection of all finite sets of evaluation points. However, this objective function is still challenging to optimize in practice: The supremum cannot be obtained analytically and the KL divergence term itself is analytically intractable and difficult to estimate in high dimensions—even for a single evaluation point.

In the next section, we will describe an approximation and estimation procedure that allows scaling function-space variational inference to large neural networks and high-dimensional input data.

# 3 Deriving a Tractable Function-Space Variational Objective

The primary obstacle to computing the objective in Equation (6) is the KL divergence from $q_{f(\cdot\,;\boldsymbol{\Theta})}$ to $p_{f(\cdot\,;\boldsymbol{\Theta})}$. There are two reasons why the KL divergence in Equation (7) is intractable: First, for BNNs or other non-linear models, we do not have access to the probability density functions of the multivariate distributions $q_{f(\mathbf{X};\boldsymbol{\Theta})}$ and $p_{f(\mathbf{X};\boldsymbol{\Theta})}$; second, for all but extremely simple input spaces, we are unable to compute the supremum over all possible finite sets of evaluation points. In the remainder of this section, we outline an approach for obtaining an estimator of a locally accurate approximation to the KL divergence that allows for scalable gradient-based optimization of Equation (7).

We first approach the problem of computing the KL divergence between two BNNs evaluated at a finite set of points. To do so, we first derive tractable approximations to the distributions over functions $q_{f(\mathbf{X};\boldsymbol{\Theta})}$ and $p_{f(\mathbf{X};\boldsymbol{\Theta})}$ Next, we show that under these approximations, we are able to obtain a closed-form approximation to the KL divergence and describe a simple Monte Carlo estimator of the supremum in the function-space KL divergence.

## 3.1 Approximating Distributions over Functions via Local Linearization

To obtain an approximation to the probability distributions of $q_{f(\mathbf{X};\boldsymbol{\Theta})}$ and $p_{f(\mathbf{X};\boldsymbol{\Theta})}$, we use a first-order Taylor expansion of the *mapping* $f$ about the mean parameters of $q_{\boldsymbol{\Theta}}$ and $p_{\boldsymbol{\Theta}}$, respectively, and derive the induced distributions under the linearized mapping.

For a stochastic function $f(\cdot\,;\boldsymbol{\Theta})$ defined in terms of stochastic parameters $\boldsymbol{\Theta}$ distributed according to distribution $g_{\boldsymbol{\Theta}}$ with $\mathbf{m} \doteq \mathbb{E}_{g_{\boldsymbol{\Theta}}}[\boldsymbol{\Theta}]$ and $\mathbf{S} \doteq \mathrm{Cov}_{g_{\boldsymbol{\Theta}}}[\boldsymbol{\Theta}]$, we denote the linearization of the stochastic function $f(\cdot\,;\boldsymbol{\Theta})$ about $\mathbf{m}$ by

$$f(\cdot\,;\boldsymbol{\Theta}) \approx \tilde{f}(\cdot\,;\mathbf{m},\boldsymbol{\Theta}) \doteq f(\cdot\,;\mathbf{m}) + \mathcal{J}(\cdot\,;\mathbf{m})(\boldsymbol{\Theta} - \mathbf{m}), \tag{8}$$

where $\mathcal{J}(\cdot\,;\mathbf{m}) \doteq (\partial f(\cdot\,;\boldsymbol{\Theta})/\partial\boldsymbol{\Theta})|_{\boldsymbol{\Theta}=\mathbf{m}}$ is the Jacobian of $f(\cdot\,;\boldsymbol{\Theta})$ evaluated at $\boldsymbol{\Theta} = \mathbf{m}$, and the mean and covariance of the distribution over the linearized mapping $\tilde{f}$ at $\mathbf{X}, \mathbf{X}' \in \mathcal{X}$ are given by

$$\mathbb{E}[\tilde{f}(\mathbf{X};\boldsymbol{\Theta})] = f(\mathbf{X};\mathbf{m}) \tag{9}$$

$$\mathrm{Cov}[\tilde{f}(\mathbf{X};\boldsymbol{\Theta}), \tilde{f}(\mathbf{X}';\boldsymbol{\Theta})] = \mathcal{J}(\mathbf{X};\mathbf{m})\mathbf{S}\mathcal{J}(\mathbf{X}',\mathbf{m})^\top. \tag{10}$$

For a derivation of this result, see Appendix A. Since Gaussianity is preserved under affine transformations, if $g_{\boldsymbol{\Theta}}$ is a multivariate Gaussian distribution with mean $\mathbf{m}$ and diagonal co-variance $\mathbf{S}$, then the distribution $\tilde{g}$ over $\tilde{f}(\mathbf{X}\,;\boldsymbol{\Theta})$ is given by

$$\tilde{g}_{\tilde{f}(\mathbf{X};\mathbf{m},\boldsymbol{\Theta})} = \mathcal{N}(f(\mathbf{X};\mathbf{m}), \mathcal{J}(\mathbf{X};\mathbf{m})\mathbf{S}\mathcal{J}(\mathbf{X};\mathbf{m})^\top). \tag{11}$$

For stochastic functions parameterized by many millions of parameters, obtaining the covariance of $\tilde{g}_{\tilde{f}(\mathbf{X};\boldsymbol{\Theta})}$—which requires computing an inner product of two Jacobian matrices—can be computationally expensive. Instead of computing the distribution over the linearized mapping exactly, we can construct a suitable Monte Carlo estimator. To do so, we consider a partition of the set of parameters into sets $\alpha$ and $\beta$ (with $|\beta| \ll |\alpha|$) and note that the linearized mapping can then be expressed as

$$\tilde{f}(\cdot\,;\mathbf{m},\boldsymbol{\Theta}) = f(\cdot\,;\mathbf{m}) + \tilde{f}_\alpha(\cdot\,;\mathbf{m},\boldsymbol{\Theta}_\alpha) + \mathcal{J}_\beta(\cdot\,;\mathbf{m})(\boldsymbol{\Theta}_\beta - \mathbf{m}_\beta), \tag{12}$$

with

$$\tilde{f}_\alpha(\cdot\,;\mathbf{m},\boldsymbol{\Theta}_\alpha) \doteq \mathcal{J}_\alpha(\cdot\,;\mathbf{m})(\boldsymbol{\Theta}_\alpha - \mathbf{m}_\alpha), \tag{13}$$

where $\mathcal{J}_\alpha(\cdot\,;\mathbf{m})$ and $\mathcal{J}_\beta(\cdot\,;\mathbf{m})$ are the columns of the Jacobian matrix corresponding to the sets of parameters $\alpha$ and $\beta$, respectively, and $\boldsymbol{\Theta}_\alpha$ and $\boldsymbol{\Theta}_\beta$ are the corresponding random parameter vectors. Noting that Equation (12) expresses $\tilde{f}$ as a sum of (affine transformations of) random variables, we can use the fact that for independent Gaussian random variables $\mathbf{X}$ and $\mathbf{Y}$, the distribution $h_{\mathbf{Z}}$ of $\mathbf{Z} = \mathbf{X} + \mathbf{Y}$ is equal to the convolution of the distributions $h_{\mathbf{X}}$ and $h_{\mathbf{Y}}$ to obtain an approximation to $\tilde{f}$. In particular, we can show that if $g_{\boldsymbol{\Theta}}$ is a multivariate Gaussian distribution with $\boldsymbol{\Theta}_\alpha \perp \boldsymbol{\Theta}_\beta$, the distribution $\tilde{g}_{\tilde{f}(\mathbf{X};\boldsymbol{\Theta})}$ can be approximated by the Monte Carlo estimator

$$\hat{\tilde{g}}_{\tilde{f}(\mathbf{X};\mathbf{m},\boldsymbol{\Theta})} = \frac{1}{R} \sum_{j=1}^R \mathcal{N}\Big(f(\mathbf{X};\mathbf{m}) + \tilde{f}_\alpha(\mathbf{X};\mathbf{m},\boldsymbol{\Theta}_\alpha)^{(j)}, \mathcal{J}_\beta(\mathbf{X};\mathbf{m})\mathbf{S}_\beta\mathcal{J}_\beta(\mathbf{X};\mathbf{m})^\top\Big), \tag{14}$$

where $g_{\boldsymbol{\Theta}_\beta} = \mathcal{N}(\mathbf{m}_\beta, \mathbf{S}_\beta)$ and samples $\tilde{f}_\alpha(\mathbf{X};\mathbf{m},\boldsymbol{\Theta}_\alpha)^{(j)}$ are obtained by sampling parameters from the distribution $g_{\boldsymbol{\Theta}_\alpha} = \mathcal{N}(\mathbf{m}_\alpha, \mathbf{S}_\alpha)$. For a derivation of this result, see Appendix A. This estimator is biased for finite $K$ but converges to $\tilde{g}_{\tilde{f}(\mathbf{X};\mathbf{m},\boldsymbol{\Theta})}$ as $R \to \infty$. Similarly, for finite $R$, the smaller $[\mathbf{S}_\alpha]_{ii}$, the more accurate and less biased the estimator will be. In our empirical evaluation, we use a single Monte Carlo sample, $R = 1$, to preserve Gaussianity and choose $\alpha$ to be the set of parameters in neural network layers $1 : L - 1$ and $\beta$ to be the set of parameters in the final neural network layer.

## 3.2 Approximating the Function-Space Kullback-Leibler Divergence

From Section 3.1, we know that if $q_\Theta$ and $p_\Theta$ are both Gaussian distributions, then the induced distributions under the linearized mapping $\tilde{f}$ evaluated at a finite set of evaluation points will be Gaussian as well. This means that for Gaussian variational and prior distributions over $\Theta$, we can obtain locally accurate approximations to the induced distributions $q_{f(\cdot;\Theta)}$ to $p_{f(\cdot;\Theta)}$ and use them to approximate the KL divergence in the variational objective by $\mathbb{D}_{\mathrm{KL}}(\tilde{q}_{\tilde{f}(\mathbf{X};\Theta)} \| \tilde{p}_{\tilde{f}(\mathbf{X};\Theta)})$. Moreover, for an isotropic Gaussian prior and a mean-field Gaussian variational distribution, $\mathbb{D}_{\mathrm{KL}}(\tilde{q}_{\tilde{f}(\mathbf{X};\Theta)} \| \tilde{p}_{\tilde{f}(\mathbf{X};\Theta)})$ is a KL divergence between two multivariate Gaussians and can be obtained analytically.

Using this approximation, we obtain an estimator of the variational objective given by

$$\tilde{\mathcal{F}}(q_\Theta) \doteq \mathbb{E}_{q_{f(\mathbf{X}_\mathcal{D};\Theta)}}[\log p_{\mathbf{y}|f(\mathbf{X};\Theta)}(\mathbf{y}_\mathcal{D} \mid f(\mathbf{X}_\mathcal{D};\boldsymbol{\theta}))] - \sup_{\mathbf{X}\in\mathcal{X}_\mathbb{N}} \mathbb{D}_{\mathrm{KL}}(\tilde{q}_{\tilde{f}(\mathbf{X};\Theta)} \| \tilde{p}_{\tilde{f}(\mathbf{X};\Theta)}), \quad (15)$$

where the arguments of the KL divergence have been replaced by the (locally accurate) approximations to the variational and prior distributions over functions evaluated at $\mathbf{X}$, respectively. Since the stochastic functions $\tilde{f}(\cdot;\Theta)$ induced by $q_\Theta$ and $p_\Theta$ under the linearized mapping will be closer to the stochastic function under $f$ the smaller the variance of $q_\Theta$ and $p_\Theta$, respectively, the approximation to the KL divergence will be more accurate the smaller the variance of $q_\Theta$ and $p_\Theta$.

Next, we turn to computing the supremum. Unlike Sun et al. [2019], who consider the supremum as a separate optimization problem, we do not seek to compute the supremum by searching over points $\mathbf{X} \in \mathcal{X}_\mathbb{N}$ but instead propose to estimate the supremum at every gradient step via a simple finite-sample estimator. Specifically, letting $I(\mathbf{X}) \doteq \mathbb{D}_{\mathrm{KL}}(\tilde{q}_{\tilde{f}(\mathbf{X};\Theta)} \| \tilde{p}_{\tilde{f}(\mathbf{X};\Theta)})$, we estimate $G = \sup_{\mathbf{X}\in\mathcal{X}_\mathbb{N}} I(\mathbf{X})$ using the Monte Carlo estimator

$$\hat{G}(\mathcal{X}_\mathcal{C}^S) = \max_{\mathbf{X}\in\mathcal{X}_\mathcal{C}^S} I(\mathbf{X}), \quad (16)$$

where $\mathcal{X}_\mathcal{C}^S \doteq \{\mathbf{X}_\mathcal{C}^{(i)}\}_{i=1}^S$ is a collection of $S$ sets of *context points* $\mathbf{X}_\mathcal{C}^{(i)} \doteq \{\mathbf{x}^{(j)}\}_{j=1}^K$ jointly sampled from a context distribution $p_{\mathcal{X}_\mathcal{C}}$. Each context set $\mathbf{X}_\mathcal{C}^{(i)}$ can be viewed as a single Monte Carlo sample from the input space so that the estimator $\hat{G}(\mathcal{X}_\mathcal{C}^S)$ provides an $S$-sample Monte Carlo estimate of the supremum. While this estimator is crude and only provides a rough approximation to the true supremum, it encourages the variational distribution over functions to match the prior distribution over functions on the sets of context points. The choice of the context distribution $p_{\mathcal{X}_\mathcal{C}}$ can be informed by knowledge about the prediction task and should be viewed as a problem-specific modeling choice. Similarly, the numbers of samples $S$ and $K$ are hyperparameters to be optimized with a validation set. For details on how $p_{\mathcal{X}_\mathcal{C}}$ is chosen for the empirical evaluation in Section 5, see Appendix D.

## 3.3 Stochastic Estimation of the Approximnate Function-Space Variational Objective

Let $q_\Theta$ be a Gaussian mean-field variational distribution, let $p_\Theta$ be an isotropic Gaussian prior, let $(\mathbf{X}_\mathcal{B}, \mathbf{y}_\mathcal{B})$ be a mini-batch of the training data, and reparameterize $\Theta$ as $\hat{\Theta}(\boldsymbol{\mu}, \boldsymbol{\Sigma}, \boldsymbol{\epsilon}^{(j)}) \doteq \boldsymbol{\mu} + \boldsymbol{\Sigma} \odot \boldsymbol{\epsilon}^{(j)}$. Using the estimator $\hat{G}(\mathcal{X}_\mathcal{C}^S)$ defined above and estimating the expected log-likelihood via Monte Carlo sampling, we obtain a Monte Carlo estimator for the function-space variational objective:

$$\bar{\mathcal{F}}(\boldsymbol{\mu}, \boldsymbol{\Sigma}) = \frac{1}{M} \sum_{j=1}^M \log p_{\mathbf{y}|f(\mathbf{X};\Theta)}(\mathbf{y}_\mathcal{B} \mid f(\mathbf{X}_\mathcal{B}; \hat{\Theta}(\boldsymbol{\mu}, \boldsymbol{\Sigma}, \boldsymbol{\epsilon}^{(j)}))) - \max_{\mathbf{X}\in\mathcal{X}_\mathcal{C}^S} \mathbb{D}_{\mathrm{KL}}(\tilde{q}_{\tilde{f}(\mathbf{X};\hat{\Theta})} \| \tilde{p}_{\tilde{f}(\mathbf{X};\hat{\Theta})})$$

$$(17)$$

with $\boldsymbol{\epsilon}^{(j)} \sim \mathcal{N}(\mathbf{0}, \mathbf{I}_P)$ and $\mathcal{X}_\mathcal{C}^S$ as defined above. This Monte Carlo estimator is biased due to the linearization and context-set approximations but allows for scalable gradient-based stochastic optimization.

**Selection of Prior.** For all experiments that involve uncertainty quantification, we chose a prior distribution over parameters that induces a prior distribution over functions $p_{f(\cdot;\Theta)}$ and a prior predictive distribution that exhibit a high degree of predictive uncertainty at evaluation points from regions in input space where $p_{\mathcal{X}_\mathcal{C}}$ has non-zero support and, under smoothness constraints, on evaluation points in nearby regions. For settings where prior information is encoded in data—for example, in the form of expert demonstrations of robotic manipulation tasks [Rudner et al., 2021] or in the form of pre-trained networks in continual or transfer learning [Rudner et al., 2022]—an empirical prior that reflects this information can be specified. For further details, see Appendix D.

**Selection of Context Distribution.** The distribution $p_{\mathcal{X}_C}$ allows us to incorporate information about the data-generating process into training and encourage the variational distribution to match the prior over functions in relevant parts of the input space. By taking advantage of the abundance of data available in real-world settings, context distributions can be constructed from large datasets like ImageNet [Krizhevsky et al., 2012], from small but diverse datasets like CIFAR-100, or by using any set of task-related unlabeled data. In our experiments, we choose two types of context distributions. One of the context distributions is constructed from the training data and only contains randomly sampled monochrome images, and one is constructed from a real-world dataset generated from a data distribution related to that of the training data. For example, when training on FashionMNIST, we use KMNIST as the context distribution, and when training on CIFAR-10, we use CIFAR-100 as the context distribution. For further details, see Appendix D.

**Posterior Predictive Distribution.** After optimizing the variational objective with respect to the parameters of the variational distribution $q_{\boldsymbol{\Theta}}$, we use the fact that we can obtain function draws by sampling from the distribution over parameters to obtain an approximate posterior predictive distribution

$$
\begin{aligned}
q(\mathbf{y}_* \,|\, \mathbf{x}_*) &= \int p(\mathbf{y}_* \,|\, f(\mathbf{x}_*; \boldsymbol{\theta}))\, q_{f(\mathbf{x}_*; \boldsymbol{\Theta})}\; \mathrm{d}f(\mathbf{x}_*; \boldsymbol{\theta}) \\
&\approx \frac{1}{M_*} \sum\nolimits_{j=1}^{M_*} p(\mathbf{y}_* \,|\, f(\mathbf{x}_*; \boldsymbol{\Theta}^{(j)})) \quad \text{with} \quad \boldsymbol{\Theta}^{(j)} \sim q_{\boldsymbol{\Theta}},
\end{aligned}
\tag{18}
$$

where $M_*$ is the number of Monte Carlo samples used to estimate the predictive distribution.

## 4 Related Work

There is a growing body of work on function-space approaches to inference in BNNs, deep learning, and applications such as continual learning [Benjamin et al., 2019, Sun et al., 2019, Titsias et al., 2020, Burt et al., 2021, Pan et al., 2020, Ma and Hernández-Lobato, 2021, Rudner et al., 2022].

**Function-Space Inference in Bayesian Neural Networks.** Previously proposed methods for FSVI in BNNs are based on approximate gradient estimators and either replace the supremum in Equation (7) with an expectation [Sun et al., 2019] or do not define an explicit variational objective [Wang et al., 2019]. Sun et al. [2019] and Carvalho et al. [2020] use Gaussian process priors over functions for which the function-space variational inference problem is not well-defined (see Section 2.1 and Burt et al. [2021]). More recent work has attempted to circumvent the intractability of the variational objective in Equation (6) by proposing alternative objectives for function-space inference in BNNs [Ma et al., 2019, Ober and Aitchison, 2020, Ma and Hernández-Lobato, 2021]. Rudner et al. [2022] extend the approach presented in Section 3 to sequential inference problems and apply it to continual learning.

**Linear Models.** Immer et al. [2020] and Khan et al. [2019] show that approximate BNN posterior distribution via the Laplace and Generalized-Gauss-Newton approximation corresponds to exact posteriors under linearizations of different models. Unlike in our approach, they use a Laplace approximation and do not perform variational inference and do not optimize the variance parameters. Furthermore, Immer et al. [2020] and Khan et al. [2019] use a neural network model to obtain a parameter maximum a posteriori estimate, but then use a linearization of the neural network model to compute a posterior predictive distribution. In contrast, our work only uses the linearization to obtain an estimator of the variational objective but uses the unlinearized model to construct a posterior predictive distribution.

**Pathologies of Variational Inference in Bayesian Neural Networks.** Burt et al. [2021] consider the function-space variational objective in Equation (6) and show that the KL divergence between BNNs with different networks architectures are not well-defined. A parallel line of research showed that posterior predictive distributions of shallow BNNs with mean-field variational distributions have a limited ability to represent complex covariance structures in function space [Foong et al., 2019, 2020] but that deep BNNs do not suffer from this limitation [Farquhar et al., 2020b]. Our results are consistent with the findings of Farquhar et al. [2020b] that mean-field variational distributions are able to represent complex covariance structures in function space.

**Table 1:** Comparison of in- and out-of-distribution performance metrics on FashionMNIST (mean $\pm$ standard error over ten random seeds). The last two columns show the AUROC for binary in- vs. out-of-distribution detection on MNIST (M) and NotMNIST (NM). MNIST and NotMNIST are used as out-of-distribution datasets. Best overall results for single and ensemble models are printed in boldface with gray shading. Results within a 95% confidence interval of the best overall result are printed in boldface only. All methods use the same four-layer CNN architecture. For further details about model architectures and training and evaluation protocols, see Appendix D.

| Method | Accuracy ↑ | ECE ↓ | AUROC M ↑ | AUROC NM ↑ |
|---|---|---|---|---|
| MAP | $91.73_{\pm0.08}$ | $0.037_{\pm0.001}$ | $87.00_{\pm0.30}$ | $74.85_{\pm1.31}$ |
| MFVI [Blundell et al., 2015] | $91.03_{\pm0.04}$ | $0.038_{\pm0.001}$ | $93.10_{\pm0.34}$ | $88.88_{\pm0.74}$ |
| MFVI (tempered) | $91.38_{\pm0.05}$ | $0.058_{\pm0.001}$ | $86.30_{\pm0.29}$ | $80.78_{\pm0.68}$ |
| MFVI (radial) [Farquhar et al., 2020a] | $90.31_{\pm0.11}$ | $0.035_{\pm0.001}$ | $84.40_{\pm0.68}$ | $82.11_{\pm1.15}$ |
| MC DROPOUT [Gal and Ghahramani, 2016] | $90.55_{\pm0.04}$ | $0.012_{\pm0.001}$ | $88.46_{\pm0.57}$ | $80.02_{\pm1.04}$ |
| SWAG [Maddox et al., 2019] | $92.56_{\pm0.05}$ | $0.043_{\pm0.001}$ | $85.18_{\pm0.35}$ | $80.31_{\pm0.30}$ |
| DUQ [van Amersfoort et al., 2020] | $92.40_{\pm0.20}$ | $-$ | $95.50_{\pm0.70}$ | $94.60_{\pm1.80}$ |
| BNN-LAPLACE [Immer et al., 2020] | $92.25_{\pm0.10}$ | $0.012_{\pm0.003}$ | $95.55_{\pm0.60}$ | $-$ |
| SPG [Ma and Hernández-Lobato, 2021] | $91.60_{\pm0.14}$ | $-$ | $95.60_{\pm6.00}$ | $-$ |
| FSVI ($p_{\mathbf{X}_\mathcal{C}}$ = random monochrome) | $93.13_{\pm0.13}$ | $0.012_{\pm0.002}$ | $96.23_{\pm0.46}$ | $95.02_{\pm0.69}$ |
| FSVI ($p_{\mathbf{X}_\mathcal{C}}$ = KMNIST) | $\mathbf{93.48}_{\pm0.12}$ | $\mathbf{0.010}_{\pm0.001}$ | $\mathbf{99.80}_{\pm0.20}$ | $\mathbf{97.26}_{\pm0.23}$ |
| Deep Ensemble | $92.49_{\pm0.01}$ | $\mathbf{0.019}_{\pm0.000}$ | $89.22_{\pm0.09}$ | $83.17_{\pm0.91}$ |
| FSVI Ensemble ($p_{\mathbf{X}_\mathcal{C}}$ = random monochrome) | $\mathbf{94.44}_{\pm0.07}$ | $0.020_{\pm0.001}$ | $\mathbf{97.85}_{\pm0.15}$ | $\mathbf{96.95}_{\pm0.20}$ |

## 5 Empirical Evaluation

In this section, we evaluate FSVI on high-dimensional classification tasks that were out of reach for function-space variational inference methods proposed in prior works and compare FSVI to several well-established and state-of-the-art Bayesian deep learning and deterministic uncertainty quantification methods. We show that FSVI (sometimes *significantly*) outperforms existing Bayesian and non-Bayesian methods in terms of their in-distribution uncertainty calibration and out-of-distribution predictive uncertainty estimation. For a details on models, training and validation procedures, and datasets used, see Appendix D. For a comparison to Sun et al. [2019] on small-scale regression tasks, see Appendix B.2.

### 5.1 Predictive Performance, Uncertainty Estimation, and Distribution Shift Detection

In this set of experiments, we assess the reliability of the uncertainty estimates generated by FSVI. If a BNN trained via FSVI is able to perform reliable uncertainty estimation, its predictive uncertainty will be significantly higher on input points that were generated according to a different data-generating distribution than the training data. For models trained on the FashionMNIST dataset, we use the MNIST and NotMNIST datasets as out-of-distribution evaluation points, while for models trained on the CIFAR-10 dataset, we use the SVHN dataset as out-of-distribution evaluation points.

For models trained on either FashionMNIST or CIFAR-10, we evaluate their in-distribution performance in terms of test accuracy, test log-likelihood, and test calibration. To evaluate the quality of different models' uncertainty estimates, we compute uncertainty estimates for the pairs FashionMNIST/MNIST, FashionMNIST/NotMNIST, and CIFAR-10/SVHN to and measure for a range of thresholds how well the datasets in each pair can be separated solely based on the uncertainty estimates. This experiment setup follows prior work by van Amersfoort et al. [2020] and Immer et al. [2020]. We report the area under the receiver operating characteristic (ROC) curve in Tables 1 and 2.

**Predictive Performance and Calibration.** To assess in-distribution predictive performance and calibration, we report the test accuracy, negative log-likelihood (NLL), and expected calibration error (ECE) for models trained on FashionMNIST and CIFAR-10 in Tables 1 and 2. On both FashionMNIST and CIFAR-10, FSVI achieves the lowest NLL and either the best or second-best predictive accuracy and ECE, respectively, across all methods. Notably, FSVI significantly outperforms SPG [Ma and Hernández-Lobato, 2021], an alternative function-space variational inference method.

**Predictive Uncertainty under Distribution Shift.** In Tables 1 and 2, we report evaluation metrics that elucidate the reliability of different methods' predictive uncertainty under distribution shift. FSVI exhibits reliable predictive uncertainty estimates that allow distinguishing between in- and out-of-distribution inputs with high accuracy. As would be expected, we observe that using context

**Table 2:** Comparison of in- and out-of-distribution performance metrics on CIFAR-10 (mean ± standard error over ten random seeds). SVHN and corrupted CIFAR-10 (C-CIFAR) are used as an out-of-distribution datasets. The penultimate column shows the AUROC for binary in- vs. out-of-distribution detection on SVHN. Best overall results for single and ensemble models are printed in boldface with gray shading. Results within a 95% confidence interval of the best overall result are printed in boldface only. All methods use a ResNet-18 architecture. For further details about model architectures and training and evaluation protocols, see Appendix D.

| Method | Accuracy↑ | ECE↓ | OOD-AUROC↑ | C-CIFAR Acc↑ |
|---|---|---|---|---|
| MAP | $93.19_{\pm0.11}$ | $0.043_{\pm0.001}$ | $94.65_{\pm0.27}$ | $78.87_{\pm1.39}$ |
| MFVI [Blundell et al., 2015] | $89.98_{\pm0.09}$ | $0.040_{\pm0.001}$ | $92.14_{\pm0.34}$ | $79.36_{\pm1.35}$ |
| MFVI (tempered) | $90.87_{\pm0.11}$ | $0.048_{\pm0.001}$ | $91.82_{\pm0.90}$ | $79.86_{\pm1.32}$ |
| MC DROPOUT [Gal and Ghahramani, 2016] | $93.55_{\pm0.07}$ | $0.040_{\pm0.001}$ | $92.44_{\pm0.57}$ | $80.13_{\pm1.37}$ |
| SWAG [Maddox et al., 2019] | $93.13_{\pm0.14}$ | $0.067_{\pm0.002}$ | $89.79_{\pm0.50}$ | $76.12_{\pm0.51}$ |
| VOGN [Osawa et al., 2019] | $84.27_{\pm0.20}$ | $0.040_{\pm0.002}$ | $87.60_{\pm0.20}$ | — |
| DUQ [van Amersfoort et al., 2020] | $94.10_{\pm0.20}$ | — | $92.70_{\pm1.30}$ | — |
| SPG [Ma and Hernández-Lobato, 2021] | $77.69_{\pm0.64}$ | — | $88.30_{\pm4.00}$ | — |
| FSVI ($p_{\mathbf{X}_\mathcal{C}}$ = random monochrome) | $93.35_{\pm0.04}$ | $0.034_{\pm0.001}$ | $94.76_{\pm0.24}$ | $\mathbf{80.81_{\pm0.43}}$ |
| FSVI ($p_{\mathbf{X}_\mathcal{C}}$ = CIFAR-100) | $93.57_{\pm0.04}$ | $\mathbf{0.026_{\pm0.001}}$ | $\mathbf{98.07_{\pm0.10}}$ | $\mathbf{81.20_{\pm0.42}}$ |
| Deep Ensemble | $95.13_{\pm0.06}$ | $0.019_{\pm0.001}$ | $98.04_{\pm0.07}$ | $81.22_{\pm0.37}$ |
| FSVI Ensemble ($p_{\mathbf{X}_\mathcal{C}}$ = random monochrome) | $\mathbf{95.19_{\pm0.03}}$ | $\mathbf{0.013_{\pm0.001}}$ | $\mathbf{99.19_{\pm0.41}}$ | $\mathbf{81.35_{\pm0.48}}$ |

distributions that reflect our knowledge about the data-generating process can significantly improve uncertainty quantification under FSVI. For the FashionMNIST experiment, we used the KMNIST dataset, which contains grayscale images of Kuzushiji letters, and for the CIFAR-10 experiment, we used the CIFAR-100 dataset, which contains RGB images of 100 classes. Both KMNIST and CIFAR-100 differ from the OOD datasets (MNIST and NotMNIST and SVHN, respectively) used to compute OOD-AUROC metrics in Tables 1 and 2, but using them as context distributions significantly increased the ability of BNNs trained via FSVI to identify distributionally shifted samples. Since the variational objective encourages matching the prior (which we chose to have high variance) on samples from the context distribution can improve uncertainty estimation in regions of the input space far from the training data.

## 5.2 Generalization and Reliability of Predictive Uncertainty under Distribution Shift

To assess the reliability of predictive models in deep learning, Ovadia et al. [2019] propose the following desiderata: In order for a model to be considered reliable, it ought to (i) exhibit low predictive uncertainty on training data and high predictive uncertainty on out-of-distribution inputs, (ii) generate predictive uncertainty estimates that allow distinguishing in- from out-of-distribution inputs, and (iii) if possible, maintain high predictive accuracy even under distribution shift. Models that satisfy these desiderata are less likely to make poor, high-confidence predictions and more amenable for use in safety-critical downstream tasks.

To illustrate these desiderata, we follow Ovadia et al. [2019] and consider the rotated MNIST task, where a model is trained on MNIST and evaluated on rotated MNIST digits. The goal is to maintain a high level of predictive accuracy (measured in terms of Brier scores) while exhibiting an increasing level of predictive uncertainty on distribution shifts of increasing magnitude. Figure 2 shows Brier scores (lower is better) and predictive entropy estimates (higher means more uncertain) of four different models. As rotating the MNIST digits gradually shifts the data distributions, we would expect Brier scores to increase (corresponding to worse predictive accuracy) as the rotation angle increases. A model with reliable predictive entropy estimates would only experience a small decrease under distribution shift while exhibiting a large increase in predictive uncertainty.

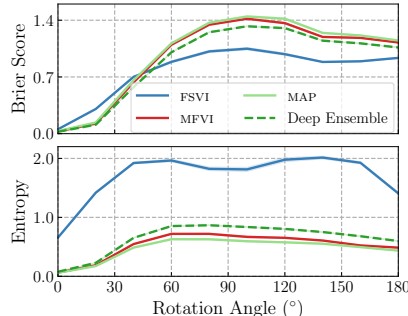

**Figure 2:** Predictive uncertainty and accuracy on rotated MNIST. Models with reliable uncertainty estimates would exhibit higher predictive uncertainty the more the digits are rotated. Ideally, such models would maintain high predictive accuracy (low Brier score).

As can be seen in the plot, the Brier scores of FSVI decreases the least, while FSVI's uncertainty is significantly higher than other models'. To assess the

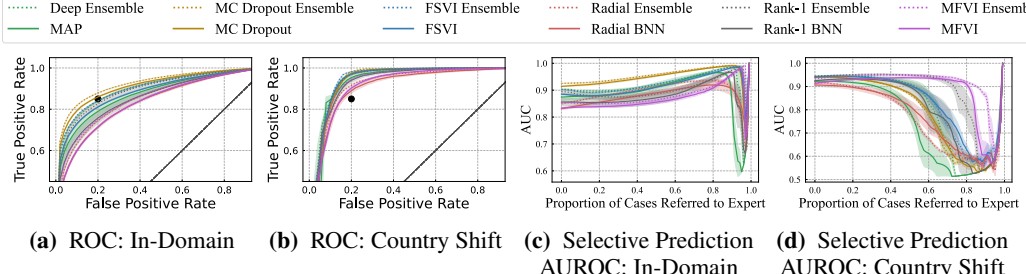

**(a)** ROC: In-Domain  **(b)** ROC: Country Shift  **(c)** Selective Prediction AUROC: In-Domain  **(d)** Selective Prediction AUROC: Country Shift

**Figure 4:** We jointly assess model predictive performance and uncertainty quantification on both in-domain and distributionally shifted data. **Left:** The *receiver operating characteristic curve* (ROC) for in-population diagnosis on the **(a)** EyePACS [2015] test set and for **(b)** changing medical equipment and patient populations on the APTOS [2019] test set. The dot in **black** denotes the NHS-recommended 85% sensitivity and 80% specificity ratios [Widdowson, 2016]. **Right:** *Selective prediction* on AUROC in **(c)** EyePACS [2015] and **(d)** APTOS [2019] settings. Shading denotes standard error over six random seeds. See Appendix B.1 for tabular results.

reliability of different uncertainty quantification methods on a more challenging distribution-shift task, we consider corrupted CIFAR-10 inputs under the second-mildest corruption level used in [Ovadia et al., 2019] and report our results in Table 2. Consistent with the rotated MNIST results, FSVI achieves the highest accuracy on the corrupted data.

### 5.3 Safety-Critical Uncertainty-Aware Selective Prediction: Diabetic Retinopathy Diagnosis

To evaluate the reliability of the predictive uncertainty of FSVI in a real-world safety-critical setting, we consider the task of diagnosing diabetic retinopathy (DR), a medical condition that can lead to impaired vision, from retina scans [Leibig et al., 2017, Filos et al., 2019, Band et al., 2021]. We use two publicly available datasets, EyePACS [2015] and APTOS [2019], each containing RGB images of a human retina graded by a medical expert on the following scale: 0 (no DR), 1 (mild DR), 2 (moderate DR), 3 (severe DR), and 4 (proliferative DR). The Kaggle dataset was collected from patients in the United States, while the APTOS dataset was collected from patients in India using cheaper but more modern scanning devices. We follow Leibig et al. [2017], Filos et al. [2019], and Band et al. [2021] and binarize all examples from both the EyePACS and AP-TOS datasets by dividing the classes up into sight-threatening diabetic retinopathy—defined as moderate diabetic retinopathy or worse (classes $\{2, 3, 4\}$)—and non-sight-threatening diabetic retinopathy—defined as no or mild diabetic retinopathy (classes $\{0, 1\}$). This results in a binary prediction task.

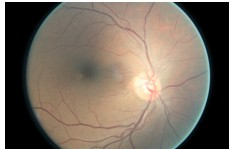
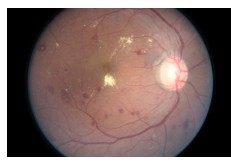

**Figure 3:** Retina scan examples. **Top:** healthy. **Bottom:** unhealthy.

To assess the reliability of predictive models when medical training and test data are obtained from different patient populations or collected with the same medical equipment, we follow Band et al. [2021] and use the Kaggle dataset for training and the distributionally shifted APTOS dataset for evaluation. The results are shown in Figure 4, which plot the ROC curves for the binary prediction problems as well as the area under the ROC curve for an uncertainty aware selective prediction task. For further details about the uncertainty-aware selective prediction evaluation protocol, see Appendix D.4. Figure 4 shows that FSVI performs well on all four tasks and is only outperformed by MC DROPOUT. For full tabular results, see Appendix B.1.

## 6 Conclusion

The paper proposed a scalable and effective approach to function-space variational inference in BNNs. We demonstrated that the proposed estimator of the function-space variational objective can be scaled up to high-dimensional data and large neural network architectures and that FSVI exhibits consistently reliable in- and out-of-distribution predictive performance on a wide range of datasets when compared to well-established and state-of-the-art uncertainty quantification methods. We hope that this work will lead to further research into function-space variational inference and the development of more sophisticated data-driven prior distributions over functions.

## Acknowledgements

We thank Bryn Elesedy, Bobby He, and Andrew Jesson for feedback on an early draft of this paper. We thank Joost van Amersfoort for helpful discussions about experiment design and implementations. Tim G. J. Rudner is funded by the Rhodes Trust and the Engineering and Physical Sciences Research Council (EPSRC). We gratefully acknowledge donations of computing resources by the Alan Turing Institute.

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
