# Supplementary Material

**Table of Contents**

# Appendix A    Proofs & Derivations

## A.1    Function-Space Variational Objective

This proof follows steps from Matthews et al. [2016]. Consider measures $\hat{P}$ and $P$ both of which define distributions over some function $f$, indexed by an infinite index set $X$. Let $\mathcal{D}$ be a dataset and let $\mathbf{X}_{\mathcal{D}}$ denote a set of inputs and $\mathbf{y}_{\mathcal{D}}$ a set of targets. Consider the measure-theoretic version of Bayes' Theorem [Schervish, 1995]:

$$\frac{d\hat{P}}{dP}(f) = \frac{p_X(Y \mid f)}{p(Y)}, \tag{A.1}$$

where $p_X(Y \mid f)$ is the likelihood and $p(Y) = \int_{\mathbb{R}^X} p_X(Y \mid f)dP(f)$ is the marginal likelihood. We assume that the likelihood function is evaluated at a finite subset of the index set $X$. Denote by $\pi_C : \mathbb{R}^X \to \mathbb{R}^C$ a projection function that takes a function and returns the same function, evaluated at a finite set of points $C$, so we can write

$$\frac{d\hat{P}}{dP}(f) = \frac{d\hat{P}_{\mathbf{X}_{\mathcal{D}}}}{dP_{\mathbf{X}_{\mathcal{D}}}}(\pi_{\mathbf{X}_{\mathcal{D}}}(f)) = \frac{p(\mathbf{y}_{\mathcal{D}} \mid \pi_{\mathbf{X}_{\mathcal{D}}}(f))}{p(\mathbf{y}_{\mathcal{D}})}, \tag{A.2}$$

and similarly, the marginal likelihood becomes $p(\mathbf{y}_{\mathcal{D}}) = \int p_{\mathbf{y}\mid f_{\mathbf{x}}}(\mathbf{y}_{\mathcal{D}} \mid f_{\mathbf{X}_{\mathcal{D}}}) \, dP_{\mathbf{X}_{\mathcal{D}}}(f_{\mathbf{X}_{\mathcal{D}}})$. Now, considering the measure-theoretic version of the KL divergence between an approximating stochastic process $Q$ and a posterior stochastic process $\hat{P}$, we can write

$$\mathbb{D}_{\mathrm{KL}}(Q \,\|\, \hat{P}) = \int \log \frac{dQ}{dP}(f) \, dQ(f) - \int \log \frac{d\hat{P}}{dP}(f) \, dQ(f), \tag{A.3}$$

where $P$ is some prior stochastic process. Now, we can apply the measure-theoretic Bayes' Theorem to obtain

$$\mathbb{D}_{\mathrm{KL}}(Q \,\|\, \hat{P}) = \int \log \frac{dQ}{dP}(f) \, dQ(f) - \int \log \frac{d\hat{P}}{dP}(f) \, dQ(f) \tag{A.4}$$

$$= \int \log \frac{dQ^\pi}{dP^\pi}(f) \, dQ^\pi(f) - \int \log \frac{d\hat{P}_{\mathbf{X}_{\mathcal{D}}}}{dP_{\mathbf{X}_{\mathcal{D}}}}(f_{\mathbf{X}_{\mathcal{D}}}) \, dQ_{\mathbf{X}_{\mathcal{D}}}(f_{\mathbf{X}_{\mathcal{D}}}) \tag{A.5}$$

$$= \int \log \frac{dQ^\pi}{dP^\pi}(f) \, dQ^\pi(f) - \mathbb{E}_{Q_{\mathbf{X}_{\mathcal{D}}}}\left[\log p\left(\mathbf{y}_{\mathcal{D}} \mid f_{\mathbf{X}_{\mathcal{D}}}\right)\right] - \log p(\mathbf{y}_{\mathcal{D}}), \tag{A.6}$$

where $\frac{dQ^\pi}{dP^\pi}(f)$ is marginally consistent given the projection $\pi$. Rearranging, we can get

$$p(\mathbf{y}_{\mathcal{D}}) = \mathbb{E}_{Q_{\mathbf{X}_{\mathcal{D}}}}\left[\log p_{\mathbf{y}\mid f_{\mathbf{x}}}(\mathbf{y}_{\mathcal{D}} \mid f_{\mathbf{X}_{\mathcal{D}}})\right] - \int \log \frac{dQ^\pi}{dP^\pi}(f) \, dQ^\pi(f) + \mathbb{D}_{\mathrm{KL}}(Q^\pi \,\|\, \hat{P}) \tag{A.7}$$

$$\geq \mathbb{E}_{Q_{\mathbf{X}_{\mathcal{D}}}}\left[\log p_{\mathbf{y}\mid f_{\mathbf{x}}}(\mathbf{y}_{\mathcal{D}} \mid f_{\mathbf{X}_{\mathcal{D}}})\right] - \int \log \frac{dQ^\pi}{dP^\pi}(f) \, dQ^\pi(f) \tag{A.8}$$

$$= \mathbb{E}_{Q_{\mathbf{X}_{\mathcal{D}}}}\left[\log p_{\mathbf{y}\mid f_{\mathbf{x}}}(\mathbf{y}_{\mathcal{D}} \mid f_{\mathbf{X}_{\mathcal{D}}})\right] - \mathbb{D}_{\mathrm{KL}}(Q^\pi \,\|\, P^\pi). \tag{A.9}$$

Finally, this lower bound can equivalently be expressed as

$$p(\mathbf{y}_{\mathcal{D}}) \geq \mathbb{E}_{Q_{\mathbf{X}_{\mathcal{D}}}}\left[\log p_{\mathbf{y}\mid f_{\mathbf{x}}}(\mathbf{y}_{\mathcal{D}} \mid f_{\mathbf{X}_{\mathcal{D}}})\right] - \mathbb{D}_{\mathrm{KL}}(Q_{\mathbf{X}_{\mathcal{D}}, \mathbf{X}_{\setminus\mathcal{D}}} \,\|\, P_{\mathbf{X}_{\mathcal{D}}, \mathbf{X}_{\setminus\mathcal{D}}}), \tag{A.10}$$

where $\mathbf{X}_{\setminus\mathcal{D}}$ is an infinite index set excluding the finite index set $\mathbf{X}_{\mathcal{D}}$, that is, $\mathbf{X}_{\setminus\mathcal{D}} \cap \mathbf{X}_{\mathcal{D}} = \varnothing$, or by Theorem 1 in Sun et al. [2019], we can write

$$p(\mathbf{y}_{\mathcal{D}}) \geq \mathbb{E}_{Q_{\mathbf{X}_{\mathcal{D}}}}\left[\log p_{\mathbf{y}\mid f_{\mathbf{x}}}(\mathbf{y}_{\mathcal{D}} \mid f_{\mathbf{X}_{\mathcal{D}}})\right] - \sup_{\mathbf{X} \in \mathcal{X}_{\mathbb{N}}} \mathbb{D}_{\mathrm{KL}}(Q_{\mathbf{X}} \,\|\, P_{\mathbf{X}}), \tag{A.11}$$

where $\mathcal{X}_{\mathbb{N}} \doteq \bigcup_{n \in \mathbb{N}}\{\mathbf{X} \in \mathcal{X}_n \mid \mathcal{X}_n \subseteq \mathbb{R}^{n \times D}\}$ is the collection of all finite sets of evaluation points.

## A.2 Distribution under Linearized Function Mapping

**Proposition 1** (Distribution under Linearized Mapping). *For a stochastic function $f(\cdot\,;\boldsymbol{\Theta})$ defined in terms of stochastic parameters $\boldsymbol{\Theta}$ distributed according to distribution $g_{\boldsymbol{\Theta}}$ with $\mathbf{m} \doteq \mathbb{E}_{g_{\boldsymbol{\Theta}}}[\boldsymbol{\Theta}]$ and $\mathbf{S} \doteq \mathrm{Cov}_{g_{\boldsymbol{\Theta}}}[\boldsymbol{\Theta}]$, denote the linearization of the stochastic function $f(\cdot\,;\boldsymbol{\Theta})$ about $\mathbf{m}$ by*

$$f(\cdot\,;\boldsymbol{\Theta}) \approx \tilde{f}(\cdot\,;\mathbf{m},\boldsymbol{\Theta}) \doteq f(\cdot\,;\mathbf{m}) + \mathcal{J}(\cdot\,;\mathbf{m})(\boldsymbol{\Theta} - \mathbf{m}),$$

*where $\mathcal{J}(\cdot\,;\mathbf{m}) \doteq (\partial f(\cdot\,;\boldsymbol{\Theta})/\partial\boldsymbol{\Theta})|_{\boldsymbol{\Theta}=\mathbf{m}}$ is the Jacobian of $f(\cdot\,;\boldsymbol{\Theta})$ evaluated at $\boldsymbol{\Theta} = \mathbf{m}$. Then the mean and co-variance of the distribution over the linearized mapping $\tilde{f}$ at $\mathbf{X}, \mathbf{X}' \in \mathcal{X}$ are given by*

$$\mathbb{E}[\tilde{f}(\mathbf{X};\boldsymbol{\Theta})] = f(\mathbf{X};\mathbf{m})$$

$$\mathrm{Cov}[\tilde{f}(\mathbf{X};\boldsymbol{\Theta}), \tilde{f}(\mathbf{X}';\boldsymbol{\Theta})] = \mathcal{J}(\mathbf{X};\mathbf{m})\mathbf{S}\mathcal{J}(\mathbf{X}';\mathbf{m})^{\top}.$$

*Proof.* We wish to find $\mathbb{E}[\tilde{f}(\mathbf{X};\mathbf{m},\boldsymbol{\Theta})]$ and

$$\mathrm{Cov}(\tilde{f}(\mathbf{X};\mathbf{m},\boldsymbol{\theta}), \tilde{f}(\mathbf{X}';\mathbf{m},\boldsymbol{\theta}))$$
$$= \mathbb{E}[(\tilde{f}(\mathbf{X};\mathbf{m},\boldsymbol{\theta}) - \mathbb{E}[\tilde{f}(\mathbf{X};\mathbf{m},\boldsymbol{\theta})])(\tilde{f}(\mathbf{X}';\mathbf{m},\boldsymbol{\theta}) - \mathbb{E}[\tilde{f}(\mathbf{X}';\mathbf{m},\boldsymbol{\theta})])^{\top}]. \tag{A.12}$$

To see that $\mathbb{E}[\tilde{f}(\mathbf{X};\mathbf{m},\boldsymbol{\theta})] = f(\mathbf{X};\mathbf{m})$, note that, by linearity of expectation, we have

$$\mathbb{E}[\tilde{f}(\mathbf{X};\mathbf{m},\boldsymbol{\theta})] = \mathbb{E}[f(\mathbf{X};\mathbf{m}) + \mathcal{J}(\mathbf{X};\mathbf{m})(\boldsymbol{\Theta} - \mathbf{m})]$$
$$= f(\mathbf{X};\mathbf{m}) + \mathcal{J}(\mathbf{X};\mathbf{m})(\mathbb{E}[\boldsymbol{\Theta}] - \mathbf{m}) = f(\mathbf{X};\mathbf{m}). \tag{A.13}$$

To see that $\mathrm{Cov}(\tilde{f}(\mathbf{X};\mathbf{m},\boldsymbol{\theta}), \tilde{f}(\mathbf{X}';\mathbf{m},\boldsymbol{\theta})) = \mathcal{J}(\mathbf{X};\mathbf{m})\mathbf{S}\mathcal{J}(\mathbf{X}';\mathbf{m})^{\top}$, note that in general, for a multivariate random variable $\mathbf{Z}$, $\mathrm{Cov}(\mathbf{Z}, \mathbf{Z}) = \mathbb{E}[\mathbf{Z}\mathbf{Z}^{\top}] + \mathbb{E}[\mathbf{Z}]\mathbb{E}[\mathbf{Z}]^{\top}$, and hence,

$$\mathrm{Cov}(\tilde{f}(\mathbf{X};\mathbf{m},\boldsymbol{\Theta}), \tilde{f}(\mathbf{X}';\mathbf{m},\boldsymbol{\Theta}))$$
$$= \mathbb{E}[\tilde{f}(\mathbf{X};\mathbf{m},\boldsymbol{\Theta})\tilde{f}(\mathbf{X}';\mathbf{m},\boldsymbol{\Theta})^{\top}] - \mathbb{E}[\tilde{f}(\mathbf{X};\mathbf{m},\boldsymbol{\Theta})]\mathbb{E}[\tilde{f}(\mathbf{X}';\mathbf{m},\boldsymbol{\Theta})]^{\top}. \tag{A.14}$$

We already know that $\mathbb{E}[\tilde{f}(\mathbf{X};\boldsymbol{\Theta})] = f(\mathbf{X};\mathbf{m})$, so we only need to find $\mathbb{E}[\tilde{f}(\mathbf{X};\boldsymbol{\Theta})\tilde{f}(\mathbf{X}';\boldsymbol{\Theta})^{\top}]$:

$$\mathbb{E}_{g_{\boldsymbol{\Theta}}}[\tilde{f}(\mathbf{X};\mathbf{m},\boldsymbol{\Theta})\tilde{f}(\mathbf{X}';\mathbf{m},\boldsymbol{\Theta})^{\top}] \tag{A.15}$$
$$= \mathbb{E}_{g_{\boldsymbol{\Theta}}}[(f(\mathbf{X};\mathbf{m}) + \mathcal{J}(\mathbf{X};\mathbf{m})(\boldsymbol{\Theta} - \mathbf{m}))(f(\mathbf{X}';\mathbf{m}) + \mathcal{J}(\mathbf{X}';\mathbf{m})(\boldsymbol{\Theta} - \mathbf{m}))^{\top}]$$

$$= \mathbb{E}_{g_{\boldsymbol{\Theta}}}[f(\mathbf{X};\mathbf{m})f(\mathbf{X}';\mathbf{m})^{\top} + (\mathcal{J}(\mathbf{X};\mathbf{m})(\boldsymbol{\Theta} - \mathbf{m}))(\mathcal{J}(\mathbf{X}';\mathbf{m})(\boldsymbol{\Theta} - \mathbf{m}))^{\top}$$
$$+ f(\mathbf{X};\mathbf{m})(\mathcal{J}(\mathbf{X}';\mathbf{m})(\boldsymbol{\Theta} - \mathbf{m}))^{\top} + \mathcal{J}(\mathbf{X};\mathbf{m})(\boldsymbol{\Theta} - \mathbf{m})f(\mathbf{X}';\mathbf{m})^{\top}] \tag{A.16}$$

$$= \mathbb{E}_{g_{\boldsymbol{\Theta}}}[f(\mathbf{X};\mathbf{m})f(\mathbf{X}';\mathbf{m})^{\top} + \mathcal{J}(\mathbf{X};\mathbf{m})(\boldsymbol{\Theta} - \mathbf{m})(\boldsymbol{\Theta} - \mathbf{m})^{\top}\mathcal{J}(\mathbf{X}';\mathbf{m})^{\top}$$
$$+ f(\mathbf{X};\mathbf{m})(\mathcal{J}(\mathbf{X}';\mathbf{m})(\boldsymbol{\Theta} - \mathbf{m}))^{\top} + \mathcal{J}(\mathbf{X};\mathbf{m})(\boldsymbol{\Theta} - \mathbf{m})f(\mathbf{X}';\mathbf{m})^{\top}] \tag{A.17}$$

$$= f(\mathbf{X};\mathbf{m})f(\mathbf{X}';\mathbf{m})^{\top} + \mathcal{J}(\mathbf{X};\mathbf{m})\mathbb{E}_{g_{\boldsymbol{\Theta}}}[(\boldsymbol{\Theta} - \mathbf{m})(\boldsymbol{\Theta} - \mathbf{m})^{\top}]\mathcal{J}(\mathbf{X}';\mathbf{m})^{\top}$$
$$+ f(\mathbf{X};\mathbf{m})(\mathcal{J}(\mathbf{X}';\mathbf{m})\underbrace{(\mathbb{E}_{g_{\boldsymbol{\Theta}}}[\boldsymbol{\Theta}] - \mathbf{m})}_{=0})^{\top} + \mathcal{J}(\mathbf{X};\mathbf{m})\underbrace{(\mathbb{E}_{g_{\boldsymbol{\Theta}}}[\boldsymbol{\Theta}] - \mathbf{m})}_{=0}f(\mathbf{X}';\mathbf{m})^{\top}, \tag{A.18}$$

where the last line follows from the definition of $g_{\boldsymbol{\Theta}}$. By definition of the covariance, we then obtain

$$\mathbb{E}_{g_{\boldsymbol{\Theta}}}[\tilde{f}(\mathbf{X};\mathbf{m},\boldsymbol{\Theta})\tilde{f}(\mathbf{X}';\mathbf{m},\boldsymbol{\Theta})^{\top}]$$
$$= f(\mathbf{X};\mathbf{m})f(\mathbf{X}';\mathbf{m})^{\top} + \mathcal{J}(\mathbf{X};\mathbf{m})\mathbb{E}_{g_{\boldsymbol{\Theta}}}[(\boldsymbol{\Theta} - \mathbf{m})(\boldsymbol{\Theta} - \mathbf{m})^{\top}]\mathcal{J}(\mathbf{X}';\mathbf{m})^{\top} \tag{A.19}$$
$$= f(\mathbf{X};\mathbf{m})f(\mathbf{X};\mathbf{m})^{\top} + \mathcal{J}(\mathbf{X};\mathbf{m})\mathrm{Cov}(\boldsymbol{\Theta})\mathcal{J}(\mathbf{X}';\mathbf{m})^{\top}. \tag{A.20}$$

With this result, we obtain the covariance function

$$\mathrm{Cov}(\tilde{f}(\mathbf{X};\mathbf{m},\boldsymbol{\Theta}), \tilde{f}(\mathbf{X}';\mathbf{m},\boldsymbol{\Theta}))$$
$$= \mathbb{E}[\tilde{f}(\mathbf{X};\mathbf{m},\boldsymbol{\Theta})\tilde{f}(\mathbf{X}';\mathbf{m},\boldsymbol{\Theta})^{\top}] - \mathbb{E}[\tilde{f}(\mathbf{X};\mathbf{m},\boldsymbol{\Theta})]\mathbb{E}[\tilde{f}(\mathbf{X}';\mathbf{m},\boldsymbol{\Theta})]^{\top} \tag{A.21}$$
$$= \mathbb{E}[\tilde{f}(\mathbf{X};\mathbf{m},\boldsymbol{\Theta})\tilde{f}(\mathbf{X}';\mathbf{m},\boldsymbol{\Theta})^{\top}] - f(\mathbf{X};\mathbf{m})f(\mathbf{X};\mathbf{m})^{\top} + \mathcal{J}(\mathbf{X};\mathbf{m})\mathrm{Cov}(\boldsymbol{\Theta})\mathcal{J}(\mathbf{X}';\mathbf{m})^{\top} \tag{A.22}$$
$$= f(\mathbf{X};\boldsymbol{\Theta})f(\mathbf{X}';\boldsymbol{\Theta})^{\top} - f(\mathbf{X};\mathbf{m})f(\mathbf{X};\mathbf{m})^{\top} + \mathcal{J}(\mathbf{X};\mathbf{m})\mathrm{Cov}\boldsymbol{\Theta}\mathcal{J}(\mathbf{X}';\mathbf{m})^{\top} \tag{A.23}$$
$$= \mathcal{J}(\mathbf{X};\mathbf{m})\mathbb{V}[\boldsymbol{\Theta}]\mathcal{J}(\mathbf{X}';\mathbf{m})^{\top}. \tag{A.24}$$

Finally, $\mathrm{Cov}(\boldsymbol{\Theta}) = \mathbf{S}$ yields $\mathrm{Cov}(\tilde{f}(\mathbf{X};\mathbf{m},\boldsymbol{\Theta}), \tilde{f}(\mathbf{X}';\mathbf{m},\boldsymbol{\Theta})) = \mathcal{J}(\mathbf{X};\mathbf{m})\mathbf{S}\mathcal{J}(\mathbf{X}';\mathbf{m})^{\top}$. This concludes the proof. $\qquad\square$

**Proposition 2** (Approximate Distribution under Linearized Mapping). *For a stochastic function $f(\cdot\,;\Theta)$ defined in terms of stochastic parameters $\Theta$ distributed according to distribution $g_\Theta = \mathcal{N}(\mathbf{m}, \mathbf{S})$, denote the linearization of the stochastic function $f(\cdot\,;\Theta)$ about $\mathbf{m}$ by*

$$f(\cdot\,;\Theta) \approx \tilde{f}(\cdot\,;\mathbf{m}, \Theta) \doteq f(\cdot\,;\mathbf{m}) + \mathcal{J}(\cdot\,;\mathbf{m})(\Theta - \mathbf{m}),$$

*where $\mathcal{J}(\cdot\,;\mathbf{m}) \doteq (\partial f(\cdot\,;\Theta)/\partial\Theta)|_{\Theta=\mathbf{m}}$ is the Jacobian of $f(\cdot\,;\Theta)$ evaluated at $\Theta = \mathbf{m}$. Then, for a partition of the set of parameters into sets $\alpha$ and $\beta$, a distribution $g_\Theta = \mathcal{N}(\mathbf{m}, \mathbf{S})$ with $\Theta_\alpha \perp \Theta_\beta$, the distribution $\tilde{g}_{\tilde{f}(\mathbf{X};\Theta)}$ can be approximated via the Monte Carlo estimator*

$$\hat{\tilde{g}}_{\tilde{f}(\mathbf{X};\mathbf{m},\Theta)} = \frac{1}{R}\sum_{j=1}^{R}\mathcal{N}\Big(f(\mathbf{X};\mathbf{m}) + \tilde{f}_\alpha(\mathbf{X};\mathbf{m},\Theta_\alpha)^{(j)}, \mathcal{J}_\beta(\mathbf{X};\mathbf{m})\mathbf{S}_\beta\mathcal{J}(\mathbf{X}';\mathbf{m})_\beta^\top\Big), \quad \text{(A.25)}$$

*where $g_{\Theta_\alpha} = \mathcal{N}(\mathbf{m}_\alpha, \mathbf{S}_\alpha)$, $g_{\Theta_\beta} = \mathcal{N}(\mathbf{m}_\beta, \mathbf{S}_\beta)$, and*

$$\tilde{f}_\alpha(\cdot\,;\mathbf{m}, \Theta_\alpha) \doteq \mathcal{J}_\alpha(\cdot\,;\mathbf{m})(\Theta_\alpha - \mathbf{m}_\alpha), \quad \text{(A.26)}$$

*with $\mathcal{J}_\alpha(\cdot\,;\mathbf{m})$ denoting the columns of the Jacobian matrix corresponding to the sets of parameters $\alpha$ and $\tilde{f}_\alpha(\mathbf{X};\mathbf{m}, \Theta_\alpha)^{(j)}$ for $j = 1, ..., R$ obtained by sampling parameters from the distribution $g_{\Theta_\alpha} = \mathcal{N}(\mathbf{m}_\alpha, \mathbf{S}_\alpha)$.*

*Proof.* Consider a partition of the set of parameters into sets $\alpha$ and $\beta$ and express the linearized mapping as

$$\tilde{f}(\cdot\,;\mathbf{m}, \Theta) = \tilde{f}_\alpha(\cdot\,;\mathbf{m}, \Theta_\alpha) + \tilde{f}_\beta(\cdot\,;\mathbf{m}, \Theta_\beta), \quad \text{(A.27)}$$

with

$$\tilde{f}_\alpha(\cdot\,;\mathbf{m}, \Theta_\alpha) \doteq \mathcal{J}_\alpha(\cdot\,;\mathbf{m})(\Theta_\alpha - \mathbf{m}_\alpha), \quad \text{(A.28)}$$

and

$$\tilde{f}_\beta(\cdot\,;\mathbf{m}, \Theta_\beta) \doteq f(\cdot\,;\mathbf{m}) + \mathcal{J}_\beta(\cdot\,;\mathbf{m})(\Theta_\beta - \mathbf{m}_\beta), \quad \text{(A.29)}$$

where $\mathcal{J}_\alpha(\cdot\,;\mathbf{m})$ and $\mathcal{J}_\beta(\cdot\,;\mathbf{m})$ are the columns of the Jacobian matrix corresponding to the sets of parameters $\alpha$ and $\beta$, respectively, and $\Theta_\alpha$ and $\Theta_\beta$ are the corresponding random parameter vectors.

Noting that Equation (A.27) expresses $\tilde{f}$ as a sum of (affine transformations of) random variables, we can use the fact that for independent Gaussian random variables $\mathbf{X}$ and $\mathbf{Y}$, the distribution $h_\mathbf{Z}$ of $\mathbf{Z} = \mathbf{X} + \mathbf{Y}$ is equal to the convolution of the distributions $h_\mathbf{X}$ and $h_\mathbf{Y}$ to obtain an approximation to $\tilde{f}$. In particular, for $\mathbf{Z} = \mathbf{X} + \mathbf{Y}$,

$$f_\mathbf{Z}(\mathbf{z}) = \int_{-\infty}^{\infty} f_\mathbf{Y}(\mathbf{z} - \mathbf{x})f_\mathbf{X}(\mathbf{x})\,d\mathbf{x}. \quad \text{(A.30)}$$

Letting $\mathbf{X} = \tilde{f}_\alpha(\mathbf{X};\mathbf{m}, \Theta_\alpha)$, $\mathbf{Y} = \tilde{f}_\beta(\mathbf{X};\mathbf{m}, \Theta_\beta)$, and $\mathbf{X} = \tilde{f}(\mathbf{X};\Theta)$, we can write

$$\tilde{g}_{\tilde{f}(\mathbf{X};\mathbf{m},\Theta)}(\tilde{f}(\mathbf{X};\mathbf{m},\boldsymbol{\theta}))$$

$$= \int_{-\infty}^{\infty} \tilde{g}_{\tilde{f}_\beta(\mathbf{X};\mathbf{m},\Theta_\beta)}(\tilde{f}(\mathbf{X};\mathbf{m},\boldsymbol{\theta}) - \tilde{f}_\alpha(\mathbf{X};\mathbf{m},\boldsymbol{\theta}_\alpha))\tilde{g}_{\tilde{f}_\alpha(\mathbf{X};\mathbf{m},\Theta_\alpha)}(\tilde{f}_\alpha(\mathbf{X};\mathbf{m},\boldsymbol{\theta}_\alpha))\,d\mathbf{X}, \quad \text{(A.31)}$$

$$= \int_{-\infty}^{\infty} \mathcal{N}(\tilde{f}(\mathbf{X};\mathbf{m},\boldsymbol{\theta})\,;\boldsymbol{\mu}(\mathbf{X},\mathbf{m},\boldsymbol{\theta}_\alpha,\tilde{f}_\alpha), \boldsymbol{\Sigma}(\mathbf{X},\mathbf{m},\boldsymbol{\theta}_\beta,\mathbf{S}_\beta))\tilde{g}_{\tilde{f}_\alpha(\mathbf{X};\mathbf{m},\Theta_\alpha)}(\tilde{f}_\alpha(\mathbf{X};\mathbf{m},\boldsymbol{\theta}_\alpha))\,d\mathbf{X}, \quad \text{(A.32)}$$

with

$$\boldsymbol{\mu}(\mathbf{X};\mathbf{m},\boldsymbol{\theta}_\alpha,\tilde{f}_\alpha) = f(\mathbf{X};\mathbf{m}) + \tilde{f}_\alpha(\mathbf{X};\mathbf{m},\boldsymbol{\theta}_\alpha) \quad \text{(A.33)}$$

and

$$\boldsymbol{\Sigma}(\mathbf{X};\mathbf{m},\mathbf{S}_\beta,\mathcal{J}_\beta) = \mathcal{J}_\beta(\mathbf{X};\mathbf{m})\mathbf{S}_\beta\mathcal{J}_\beta(\mathbf{X};\mathbf{m})^\top, \quad \text{(A.34)}$$

where we have used the fact that for a Gaussian distribution with mean $m$ and covariance $S$, $\mathcal{N}(z - y; m, S) = \mathcal{N}(z; m + y, S)$. We can then approximate the probability density function $\tilde{g}_{\tilde{f}(\mathbf{X};\mathbf{m},\Theta)}(\tilde{f}(\mathbf{X};\boldsymbol{\theta}))$ via the Monte Carlo estimator

$$\hat{\tilde{g}}_{\tilde{f}(\mathbf{X};\mathbf{m},\Theta)}(\tilde{f}(\mathbf{X};\mathbf{m},\boldsymbol{\theta}))$$

$$= \frac{1}{R}\sum_{j=1}^{R}\mathcal{N}(\tilde{f}(\mathbf{X};\mathbf{m},\boldsymbol{\theta})\,;\boldsymbol{\mu}(\mathbf{X},\mathbf{m},\tilde{f}_\alpha(\mathbf{X};\mathbf{m},\boldsymbol{\theta}_\alpha)^{(j)}), \boldsymbol{\Sigma}(\mathbf{X};\mathbf{m},\mathbf{S}_\beta,\mathcal{J}_\beta)) \quad \text{(A.35)}$$

with $\tilde{f}_\alpha(\mathbf{X};\mathbf{m},\boldsymbol{\theta}_\alpha)^{(j)} \sim \tilde{g}_{\tilde{f}_\alpha(\mathbf{X};\mathbf{m},\Theta_\alpha)}$. Finally, we can express the distribution $\hat{\tilde{g}}_{\tilde{f}(\mathbf{X};\mathbf{m},\Theta)}$ as

$$\hat{\tilde{g}}_{\tilde{f}(\mathbf{X};\mathbf{m},\Theta)} = \frac{1}{R}\sum_{j=1}^{R}\mathcal{N}\Big(f(\mathbf{X};\mathbf{m}) + \tilde{f}_\alpha(\mathbf{X};\mathbf{m},\Theta_\alpha)^{(j)}, \mathcal{J}_\beta(\mathbf{X};\mathbf{m})\mathbf{S}_\beta\mathcal{J}_\beta(\mathbf{X};\mathbf{m})^\top\Big), \quad \text{(A.36)}$$

where $g_{\Theta_\beta} = \mathcal{N}(\mathbf{m}_\beta, \mathbf{S}_\beta)$ and samples $\tilde{f}_\alpha(\mathbf{X};\mathbf{m},\Theta_\alpha)^{(j)}$ are obtained by sampling parameters from the distribution $g_{\Theta_\alpha} = \mathcal{N}(\mathbf{m}_\alpha, \mathbf{S}_\alpha)$. This concludes the proof. □

# Appendix B  Further Empirical Results

## B.1  Tabular Results for Diabetic Retinopathy Diagnosis Tasks

The results below were reproduced from Band et al. [2021] using the RETINA benchmark.

**Table 3: Country Shift.** Prediction and uncertainty quality of baseline methods in terms of the area under the receiver operating characteristic curve (AUC) and classification accuracy, as a function of the proportion of data referred to a medical expert. All methods are tuned on in-domain validation AUC, and ensembles have $K = 3$ constituent models (true for all subsequent tables unless specified otherwise). On in-domain data, MC DROPOUT performs best across all thresholds. On distributionally shifted data, no method consistently performs best.

| Method | No Referral AUC (%) ↑ | No Referral Accuracy (%) ↑ | 50% Data Referred AUC (%) ↑ | 50% Data Referred Accuracy (%) ↑ | 70% Data Referred AUC (%) ↑ | 70% Data Referred Accuracy ↑ |
|---|---|---|---|---|---|---|
| | | | EyePACS Dataset (In-Domain) | | | |
| MAP (Deterministic) | $87.4_{\pm1.3}$ | $88.6_{\pm0.7}$ | $91.1_{\pm1.8}$ | $95.9_{\pm0.4}$ | $94.9_{\pm1.1}$ | $96.5_{\pm0.3}$ |
| MFVI | $83.3_{\pm0.2}$ | $85.7_{\pm0.1}$ | $85.5_{\pm0.7}$ | $94.5_{\pm0.1}$ | $88.2_{\pm0.7}$ | $95.9_{\pm0.1}$ |
| RADIAL-MFVI | $83.2_{\pm0.5}$ | $74.2_{\pm5.0}$ | $88.9_{\pm0.9}$ | $81.8_{\pm6.0}$ | $91.2_{\pm1.3}$ | $83.8_{\pm5.5}$ |
| FSVI | $88.5_{\pm0.1}$ | $89.8_{\pm0.0}$ | $91.0_{\pm0.4}$ | $96.4_{\pm0.0}$ | $94.3_{\pm0.3}$ | $97.2_{\pm0.1}$ |
| MC DROPOUT | $91.4_{\pm0.2}$ | $90.9_{\pm0.1}$ | $95.3_{\pm0.2}$ | $97.4_{\pm0.1}$ | $97.4_{\pm0.1}$ | $98.1_{\pm0.0}$ |
| RANK-1 | $85.6_{\pm1.4}$ | $87.7_{\pm0.8}$ | $87.1_{\pm2.3}$ | $95.3_{\pm0.5}$ | $90.9_{\pm2.0}$ | $96.4_{\pm0.4}$ |
| DEEP ENSEMBLE | $90.3_{\pm0.2}$ | $90.3_{\pm0.3}$ | $91.7_{\pm0.6}$ | $97.2_{\pm0.0}$ | $95.0_{\pm0.5}$ | $97.9_{\pm0.0}$ |
| MFVI ENSEMBLE | $85.4_{\pm0.0}$ | $87.8_{\pm0.0}$ | $86.3_{\pm0.4}$ | $95.4_{\pm0.0}$ | $89.2_{\pm0.4}$ | $96.7_{\pm0.1}$ |
| RADIAL-MFVI ENSEMBLE | $84.9_{\pm0.1}$ | $74.2_{\pm1.5}$ | $91.4_{\pm0.2}$ | $83.4_{\pm1.7}$ | $93.3_{\pm0.3}$ | $85.9_{\pm1.6}$ |
| FSVI ENSEMBLE | $90.3_{\pm0.1}$ | $90.6_{\pm0.0}$ | $92.1_{\pm0.2}$ | $97.1_{\pm0.0}$ | $95.2_{\pm0.2}$ | $97.8_{\pm0.1}$ |
| MC DROPOUT ENSEMBLE | $\mathbf{92.5_{\pm0.0}}$ | $\mathbf{91.6_{\pm0.0}}$ | $\mathbf{95.8_{\pm0.1}}$ | $\mathbf{97.8_{\pm0.0}}$ | $\mathbf{97.7_{\pm0.1}}$ | $\mathbf{98.4_{\pm0.0}}$ |
| RANK-1 ENSEMBLE | $89.5_{\pm0.8}$ | $89.3_{\pm0.4}$ | $88.5_{\pm1.3}$ | $96.9_{\pm0.3}$ | $91.6_{\pm1.2}$ | $97.6_{\pm0.3}$ |
| | | | APTOS 2019 Dataset (Population Shift) | | | |
| MAP (Deterministic) | $92.2_{\pm0.2}$ | $86.2_{\pm0.6}$ | $80.1_{\pm3.6}$ | $87.6_{\pm1.5}$ | $55.4_{\pm4.3}$ | $85.4_{\pm1.2}$ |
| MFVI | $91.4_{\pm0.2}$ | $84.1_{\pm0.3}$ | $93.8_{\pm0.4}$ | $92.1_{\pm0.5}$ | $93.0_{\pm0.6}$ | $92.7_{\pm0.5}$ |
| RADIAL-MFVI | $90.7_{\pm0.7}$ | $71.8_{\pm4.6}$ | $82.0_{\pm2.5}$ | $81.5_{\pm2.7}$ | $66.4_{\pm2.1}$ | $85.9_{\pm1.0}$ |
| FSVI | $94.1_{\pm0.1}$ | $87.6_{\pm0.5}$ | $90.6_{\pm0.9}$ | $90.7_{\pm0.7}$ | $77.2_{\pm4.6}$ | $89.8_{\pm0.3}$ |
| MC DROPOUT | $94.0_{\pm0.2}$ | $86.8_{\pm0.2}$ | $87.4_{\pm0.3}$ | $88.1_{\pm0.2}$ | $65.3_{\pm1.7}$ | $88.2_{\pm0.4}$ |
| RANK-1 | $92.5_{\pm0.3}$ | $86.2_{\pm0.5}$ | $90.1_{\pm2.5}$ | $91.4_{\pm1.1}$ | $75.1_{\pm7.8}$ | $89.5_{\pm1.5}$ |
| DEEP ENSEMBLE | $94.2_{\pm0.2}$ | $87.5_{\pm0.1}$ | $91.2_{\pm1.9}$ | $92.4_{\pm0.9}$ | $67.4_{\pm7.3}$ | $90.1_{\pm1.2}$ |
| MFVI ENSEMBLE | $93.2_{\pm0.1}$ | $87.0_{\pm0.2}$ | $\mathbf{94.9_{\pm0.3}}$ | $\mathbf{93.7_{\pm0.3}}$ | $\mathbf{94.2_{\pm0.3}}$ | $\mathbf{94.0_{\pm0.3}}$ |
| RADIAL-MFVI ENSEMBLE | $91.8_{\pm0.2}$ | $69.0_{\pm1.9}$ | $78.6_{\pm0.6}$ | $79.8_{\pm0.9}$ | $60.9_{\pm0.3}$ | $86.7_{\pm0.2}$ |
| FSVI ENSEMBLE | $\mathbf{94.6_{\pm0.1}}$ | $\mathbf{88.9_{\pm0.2}}$ | $90.7_{\pm0.5}$ | $91.1_{\pm0.6}$ | $74.1_{\pm3.4}$ | $89.8_{\pm0.2}$ |
| MC DROPOUT ENSEMBLE | $94.1_{\pm0.1}$ | $87.6_{\pm0.1}$ | $86.8_{\pm0.2}$ | $88.0_{\pm0.2}$ | $62.3_{\pm0.4}$ | $87.7_{\pm0.2}$ |
| RANK-1 ENSEMBLE | $94.1_{\pm0.2}$ | $88.3_{\pm0.2}$ | $\mathbf{94.9_{\pm0.4}}$ | $93.5_{\pm0.3}$ | $92.4_{\pm1.5}$ | $93.8_{\pm0.3}$ |

## B.2  UCI Regression

**Table 4:** This table compares the predictive performance between the method proposed in this paper and the method proposed by Sun et al. [2019] on six datasets from the UCI database. We followed the same training protocol as Sun et al. [2019] and used the code provided by the authors to load and process the data. The same network architecture was used (one hidden layer with 50 hidden units). We report the results for the best set of hyperparameters, computed over ten random seeds. Lower RMSE and higher log-likelihood are better. Best results are shaded in gray. The first five rows are small-scale UCI experiments, and the sixth row ("Protein") is a larger-scale experiment (45,740 data points).

| | RMSE Sun et al. [2019] | RMSE Ours | Log-Likelihood Sun et al. [2019] | Log-Likelihood Ours |
|---|---|---|---|---|
| Boston | $\mathbf{2.378 \pm 0.104}$ | $3.632 \pm 0.515$ | $\mathbf{-2.301 \pm 0.038}$ | $-3.150 \pm 0.495$ |
| Concrete | $4.935 \pm 0.180$ | $\mathbf{4.177 \pm 0.443}$ | $-3.096 \pm 0.016$ | $\mathbf{-2.855 \pm 0.116}$ |
| Energy | $0.412 \pm 0.017$ | $\mathbf{0.409 \pm 0.060}$ | $-0.684 \pm 0.020$ | $\mathbf{-0.539 \pm 0.138}$ |
| Wine | $0.673 \pm 0.014$ | $\mathbf{0.615 \pm 0.033}$ | $-1.040 \pm 0.013$ | $\mathbf{-0.959 \pm 0.034}$ |
| Yacht | $0.607 \pm 0.068$ | $\mathbf{0.514 \pm 0.242}$ | $-1.033 \pm 0.033$ | $\mathbf{-0.888 \pm 0.334}$ |
| Protein | $4.326 \pm 0.019$ | $\mathbf{4.248 \pm 0.043}$ | $-2.892 \pm 0.004$ | $\mathbf{-2.866 \pm 0.009}$ |

# Appendix C   Illustrative Examples

## C.1   Two Moons Classification Task

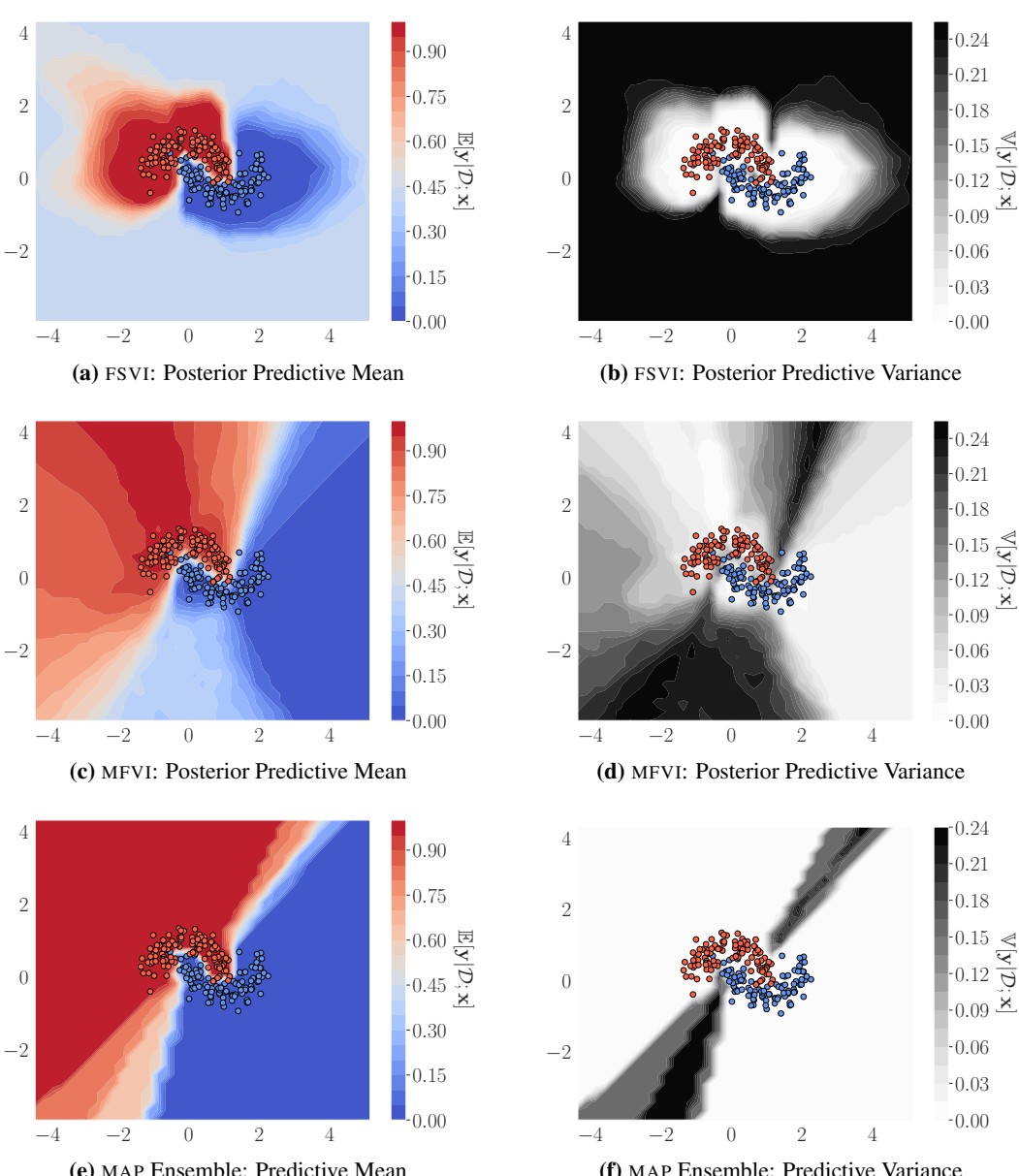

**(a)** FSVI: Posterior Predictive Mean

**(b)** FSVI: Posterior Predictive Variance

**(c)** MFVI: Posterior Predictive Mean

**(d)** MFVI: Posterior Predictive Variance

**(e)** MAP Ensemble: Predictive Mean

**(f)** MAP Ensemble: Predictive Variance

**Figure 5:** Binary classification on the *Two Moons* dataset. The plots show the posterior predictive mean and variance of a BNN trained via FSVI (Figure 5a and Figure 5b), of a BNN trained via MFVI (Figure 5c and Figure 5d), and an ensemble of MAP models (Figure 5e and Figure 5f). The predictive means represent the expected class probabilities and the predictive variance the model's epistemic uncertainty over the class probabilities. With FSVI, the predictive distribution is able to faithfully capture the geometry of the data manifold and exhibits high uncertainty over the class probabilities in areas of the data space of which the data is not informative. In contrast, neither MFVI, nor MAP ensembles are unable to accurately capture the geometry of the data manifold only exhibit high uncertainty around the decision boundary.

## C.2 Synthetic 1D Regression Datasets

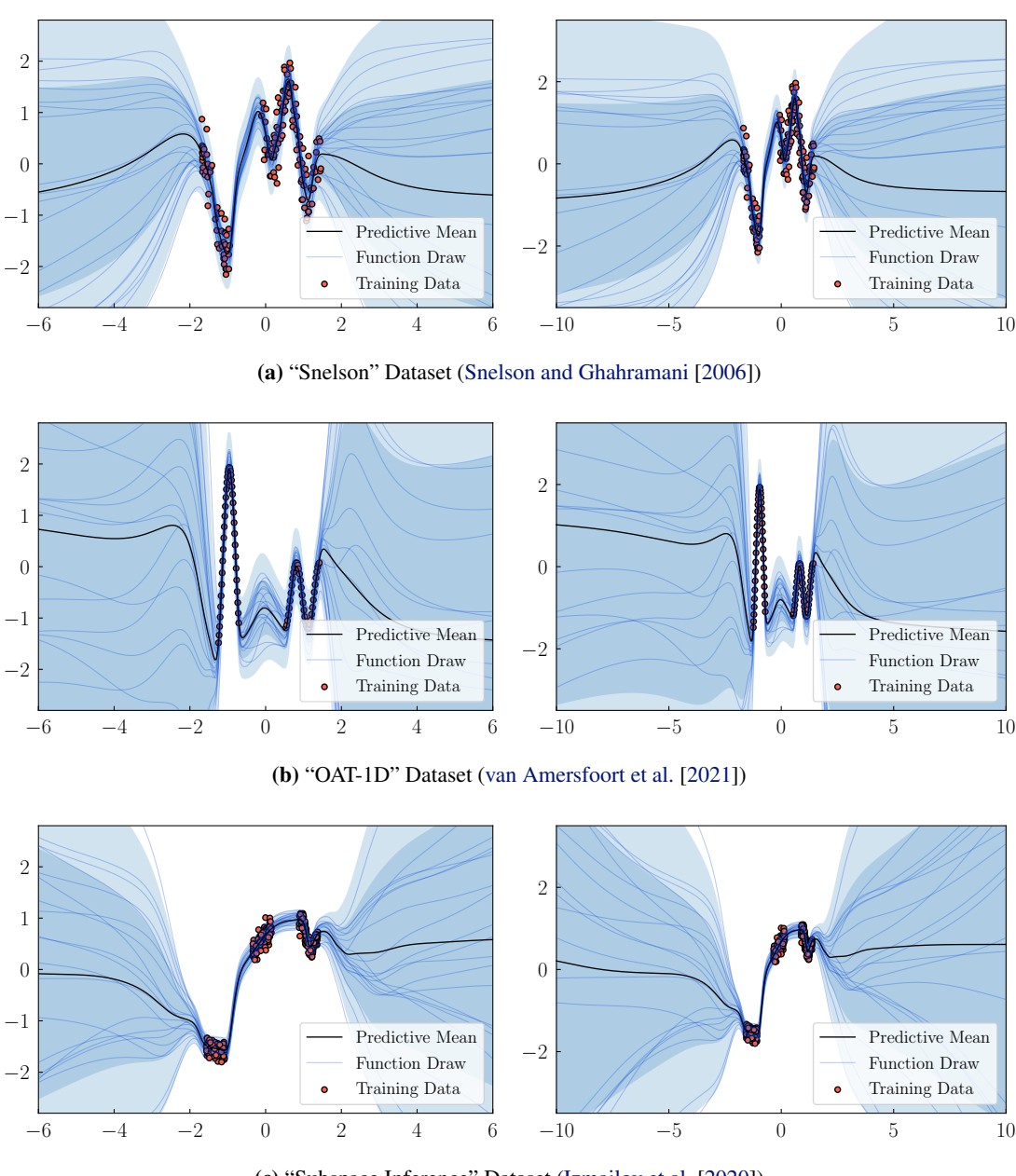

**(a)** "Snelson" Dataset (Snelson and Ghahramani [2006])

**(b)** "OAT-1D" Dataset (van Amersfoort et al. [2021])

**(c)** "Subspace Inference" Dataset (Izmailov et al. [2020])

**Figure 6:** 1D Regression with FSVI on a selection of datasets used to demonstrate desirable predictive uncertainty estimates in prior works. The left column is zoomed in.

# Appendix D    Implementation, Training, and Evaluation Details

## D.1    Hyperparameter Selection Protocol

For FSVI, we used a holdout validation set (10% of the training set) to conduct a hyperparameter search over the prior variance, the number of context points used to evaluate the KL divergence, the context distribution, and the number of Monte Carlo samples used to evaluate the expected log-likelihood. We selected the set of hyperparameters that yielded the highest validation log-likelihood for all experiments. We state the hyperparameters selected for the different datasets below.

For other methods, we used a holdout validation set of the same size and selected the best-performing hyperparameters. We used implementations provided by the authors of MFVI (radial) and SWAG. All other methods were implemented from scratch unless stated otherwise.

## D.2    FashionMNIST vs. MNIST/NotMNIST

We train all model on the FashionMNIST dataset and evaluate the models' predictive uncertainty performance on out-of-distribution data on the MNIST dataset. Both datasets consist of images of size $28 \times 28$ pixels. The FashionMNIST dataset is normalized to have zero mean and a standard deviation of one. The MNIST dataset is normalized with the same transformation, that is, using the same mean and standard deviation used for the in-distribution data. We chose FashionMNIST/MNIST instead of MNIST/NotMNIST because the latter is notably easier than the former.

In this experiment, a network architecture with two convolutional layers of 32 and 64 $3 \times 3$ filters and a fully-connected final layer of 128 hidden units is used. A max pooling operation is placed after each convolutional layer and ReLU activations are used. We do not use batch normalization. All models are trained for 30 epochs with a mini-batch size of 128 using SGD with a learning rate of $5 \times 10^{-3}$, momentum (with momentum parameter 0.9), and a cosine learning rate schedule with parameter 0.05.

For FSVI with $p_{\mathbf{X}_c} =$ random monochrome, we sampled 50% of the context points for each gradient step from the mini-batch and the other 50% according to the method described in Appendix D.8. For FSVI with $p_{\mathbf{X}_c} =$ KNIST, we used the KMNIST dataset.

## D.3    CIFAR-10 vs. SVHN

We train all model on the CIFAR-10 dataset and evaluate the models' predictive uncertainty performance on out-of-distribution data on the SVHN dataset. Both datasets consist of images of size $32 \times 32 \times 3$, with RBG channels. The CIFAR-10 dataset is normalized to have zero mean and a standard deviation of one. The SVHN dataset is normalized with the same transformation, that is, using the same mean and standard deviation used for the in-distribution data. The training data is augmented with random horizontal flips (with a probability of 0.5) and random crops (4 zero pixels on all sides).

In this experiment, a standard ResNet-18 network architecture was used. All models are trained for 200 epochs with a mini-batch size of 128 using SGD with a learning rate of $5 \times 10^{-3}$, momentum (with momentum parameter 0.9), and a cosine learning rate schedule with parameter 0.05.

For FSVI with $p_{\mathbf{X}_c} =$ random monochrome, we sampled 100% of the context points for each gradient step from the mini-batch and the other 50% according to the method described in Appendix D.8. For FSVI with $p_{\mathbf{X}_c} =$ CIFAR-100, we used the CIFAR-100 dataset.

## D.4    Diabetic Retinopathy Diagnosis

**Prediction and Expert Referral.**    In real-world settings where the evaluation data may be sampled from a shifted distribution, incorrect predictions may become increasingly likely. To account for that possibility, predictive uncertainty estimates can be used to identify datapoints where the likelihood of an incorrect prediction is particularly high and refer them for further review. We consider a corresponding selective prediction task, where the predictive performance of a given model is evaluated for varying expert referral rates. That is, for a given referral rate of $\gamma \in [0, 1]$, a model's predictive uncertainty is used to identify the $\gamma$ proportion of images in the evaluation set for which the model's predictions are most uncertain. Those images are referred to a medical professional for further

review, and the model is assessed on its predictions on the remaining $(1-\gamma)$ proportion of images. By repeating this process for all possible referral rates and assessing the model's predictive performance on the retained images, we estimate how reliable it would be in a safety-critical downstream task, where predictive uncertainty estimates are used in conjunction with human expertise to avoid harmful predictions. Importantly, selective prediction tolerates out-of-distribution examples. For example, even if unfamiliar features appear in certain images, a model with reliable uncertainty estimates will perform better in selective prediction by assigning these images high epistemic (and predictive) uncertainty, therefore referring them to an expert at a lower $\gamma$.

For all methods, experiments are performed using a ResNet-50 network architecture. Training and evaluation scripts as well as model checkpoints can be found at



`github.com/google/uncertainty-baselines/.../diabetic_retinopathy_detection`.



### D.5 Two Moons

In this experiment, we use a multi-layer perceptron (MLP) consisting of two fully-connected layers with 30 hidden units each and tanh activations. We train all models with a learning rate of $10^{-3}$.

For FSVI, we sampled context points uniformly from $[-10, 10] \times [-10, 10]$.

### D.6 1D Regression

In this experiment, we use a multi-layer perceptron (MLP) consisting of two fully-connected layers with 100 hidden units each and ReLU activations.

For FSVI, we sampled context points uniformly from $[-10, 10]$.

### D.7 Further Implementation Details

We use the Adam optimizer with default settings of $\beta_1 = 0.9$, $\beta_2 = 0.99$ and $\epsilon = 10^{-8}$ for all experiments. The deterministic neural networks that were used for the ensemble were trained with a weight decay of $\lambda = 1\text{e-}1$. MFVI (tempered) was trained with a KL scaling factor of 0.1 to obtain a cold posterior.

### D.8 Selection of Context Distribution

We estimate the supremum at every gradient step by sampling a set of context points $\mathbf{X}_{\mathcal{C}}$ from a distribution $p_{\mathbf{X}_c}$ at every gradient step. For tasks with image inputs, we construct a distribution $p_{\mathbf{X}_c}$, defined as a uniform distribution over images with monochromatic channels. To generate a sample from this "monochrome images" distribution, we first take all images in the training data, flatten each channel, and stack the flattened image channels into a single vector each. We then draw a random element (i.e., a pixel) from each channel vector and then use these pixels to generate a monochrome image of a given resolution by setting every channel equal to the value of the pixel that was drawn. For regression tasks with a $D$-dimensional input space, $p_{\mathbf{X}_c}$ is defined as a uniform distribution with lower and upper bounds set to the empirical lower and upper bounds of the training data. For further details on the effect of different sampling schemes on the posterior predictive distribution's performance, see Appendix B.

### D.9 Compute Resources

All experiments were carried out on an Nvidia V-100 GPU with 32GB of memory.