# OpenReview forum: "Tractable Function-Space Variational Inference in Bayesian Neural Networks"
_NeurIPS.cc/2022/Conference — NeurIPS 2022 Accept_

### Official Review · Reviewer_8CnJ · 2022-07-11

**Rating:** 6
**Confidence:** 4
**Soundness:** 4 excellent
**Presentation:** 4 excellent
**Contribution:** 3 good

**Summary:**

This works presents the function-space variational inference in tractable manner. Specifically, the existing function-space variational inference [1] has computational issue in computing KL terms between variational and prior distributions over function-space. This work first presents the tractable form of KL terms by approximating the distributions over function-space linearly by the Taylor approximation, and then employs the estimate of supreme of KL terms by using the context sets, which results in the function-space variational inference in tractable manner. Through experiments, authors validate that the proposed inference could perform the reliable uncertainly estimation, and outperform the existing Bayesian deep learning model for predictive performance and uncertainty estimation.

[1] Functional variational bayesian neural networks, ICLR 19

**Questions:**

$\bullet $ Q1. could you explain some intuition on why authors choose such context distributions in Table 1 and 2? what if the KMNIST or CIFAR100 is not available, which datasets do the authors use? Please explain why authors chose such datasets.

**Limitations:**

See section Weakness above.

**Strengths And Weaknesses:**

$\textbf{Strengths}$

$\bullet$ This work proposes the tractable function-space variational inference, which can easily be computed.

$\bullet$ This work has been sufficiently validated that the proposed inferences leads to the reliable uncertainly estimation and superior predictive performance on a variety of the datasets.

$\textbf{Weaknesses}$

$\bullet$ The technical novelty looks incremental.

$\bullet$ As described in methodology section and shown in Table 1 and 2, the predictive performances seem to be largely affected depending on how to chose the context points and distribution. Thus, the performance could be unstable depending on users who might have deep understanding of given task or not. Thus, I think that explaining the intuition on how to choose context distribution should be described in detail.

---

> ### Author Response · Authors · 2022-08-02
> **Response to Reviewer 8CnJ**
>
> ### General Comments
>
> Thank you for your positive review and for your questions. We were pleased to read that you found that our paper “sufficiently validated that the proposed inference [method] leads to reliable uncertainty estimation and superior predictive performance”.
>
> We hope our clarifications below address your questions and any remaining concerns and would be more than happy to answer any follow-up questions you may have.
>
>
> ### 1. Technical novelty
>
> **Quote from review:**
> > The technical novelty looks incremental.
>
> To address this point, it would be useful if you could clarify in which way you find that the proposed method appears to be incremental.
>
> We propose a conceptually simple and computationally tractable approach to a problem that the Bayesian deep learning community has worked on for several years. While lineariations are simple tools and have been explored by the Bayesian deep learning community (for example, in the context of Gauss-Newton approximations), our paper is the first to demonstrate how we can use **repeated** linearization  to obtain analytically tractable distributions over functions. In addition, we note---unlike Sun et al. (2019)---we explain how to obtain meaningful prior distributions over functions that lead to well-defined variational objective functions by showing show to effectively construct useful prior distributions over functions from pretrained models (in the continual learning setting) and suitably defined context distributions without relying on non-parametric GP priors for which the function-space KL is not well-defined (also see lines 89-97).
>
> ### 2. Impact of context distribution
>
> **Quote from review:**
> > As described in methodology section and shown in Table 1 and 2, the predictive performances seem to be largely affected depending on how to chose the context points and distribution. Thus, the performance could be unstable depending on users who might have deep understanding of given task or not. Thus, I think that explaining the intuition on how to choose context distribution should be described in detail.
>
> As described in the paper, FSVI is particularly useful when prior information, about the latent functions or about the context distribution is available. That being said, a conceptual shortcoming of function-space variational inference as described in Sun et al. (2019) is that the the input points are not viewed as part of the probabilistic model, and as such, the context distribution in FSVI should be viewed as a kind of tunable hyperparameter (as opposed to a prior in the Bayesian sense).
>
> Importantly, however, as the results in Tables 1 and 2 show, FSVI can be very effective even without deep understanding of the task at hand: Even when using the naively constructed “random monochrome” context distributions, which only contain monochrome pictures randomly generated from the distribution of pixels in the training distribution, FSVI outperforms most alternative methods in terms of predictive performance and uncertainty quantification.
>
> **Quote from review:**
> > Q1. could you explain some intuition on why authors choose such context distributions in Table 1 and 2? what if the KMNIST or CIFAR100 is not available, which datasets do the authors use? Please explain why authors chose such datasets.
>
> To better address your question and complement the results in the paper, we conducted a sensitivity analysis on the effect of different context distributions for the FashionMNIST prediction problem.
>
> The results of our experiments can be found here: [fully-anonymized link to interactive results dashboard (FashionMNIST)](https://rebrand.ly/sensitivity_analysis_context_distribution_fmnist).
>
> The results show that some distributions are more suitable than others, but we note that this is merely a “toy” setting. In real-world problems, data scientists and machine learning engineers typically have deep knowledge of the task at hand, and there are a plethora of applications, where only a small labeled dataset but large amounts of related unlabeled data is available. Such unlabeled data, especially if it might feature covariate shifts, would be particularly well-suited for constructing a context distribution. For example, our experiments with RGB images specifically use CIFAR-100 to construct a context distribution, since CIFAR-100 contains a relatively large number of classes, but other datasets, such as ImageNet, or data scraped from the internet, provide vastly larger collections of classes and, we believe, would be well-suited for many real-world image classification problems with FSVI.

---

> > ### Comment · Reviewer_8CnJ · 2022-08-06
> > **Thank you for reply and clarification on 1 Technical novelty.**
> >
> >
> > ### More clarification on technical novelty.
> >
> > First of all, I apologized for my comment for technical novelty without explaining why I thought in this way, and thus add my thought on this for clarification.
> >
> >  I believed that the technically different part of this work compared to the FSVI [1] is to make KL term between the distribution of the function $\text{KL}(q(f)||p(f))$  tractable by approximating  $q(f)$ and $p(f)$ linearly (Gauss-Newton approximations) and thus making them as Gaussian distribution. To do this, this work employs the Gauss-Newton approximations, that was developed in [2], which is not developed for this work. Also, this work did not concern much about how to estimate the supreme estimation efficiently for computing $\text{KL}(q(f)||p(f))$, and seems that this work uses the similar estimation method described in sampling-based measurement sets in [1] (I recognize that this work is distinct to [1] in that (1) $\text{KL}(q(f)||p(f))$ is analytically computed, and (2) context set are sampled from the large datasets which are different to training datasets). These are some reasons why I thought that technical novelty looks incremental.
> >
> >
> > [1] Functional variational bayesian neural networks, ICLR 19
> >
> > [2] Improving predictions of bayesian neural networks via local linearization, AISTATS 2021
> >
> >
> >
> > ### My response after reading the reply.
> >
> > Thanks for your response, and it helps me understand more about the role of the context set. Personally, I think that this work will shin even more if new technical contributions to improve functional variation BNNs, such as the KL term estimation for handing the supreme or elaborate sampling method for context set, are introduced. However, this is not intended to deliberately undermine this work. I admit that this work presents the simple way about how to obtain induce prior distributions over functions, and enables the functional BNN to be trained easily with the analytical loss. Also, the performances of this work are impressive compared to other baseline BNN. Thus, I maintain my score and support the weak acceptance.

---

> > > ### Author Response · Authors · 2022-08-09
> > > **Clarification & Correction**
> > >
> > > ### Thank you for your reply!
> > >
> > > We greatly appreciate the clarification and would like to address a misunderstanding/misconception expressed in your comment.
> > >
> > > 1. **The approximation used in our manuscript is not a Gauss-Newton approximation** and
> > >
> > > 2. we note that **the approach presented in our manuscript is distinctly different from the approach considered in Immer et al. (2021)** [1].
> > >
> > > ### 1. What's the difference between the Generalized Gauss-Newton (GGN) approximation and the proposed approximation?
> > >
> > > First, what is the GGN approximation? In the context of non-linear loss functions, the GGN approximation is an approximation to the *Hessian of a loss function* (i.e., it implicitly changes the loss function) [1]. Equivalently, the GGN approximation can be interpreted as the unchanged loss under a different *model*, as noted by [2].
> > >
> > > To see that the proposed approach does not use the GGN approximation, note that **the approximation proposed in the manuscript does not change the model**. As noted in the equation between lines 174 and 175, where the expected log-likelihood is computed under $f$ and not $\tilde{f}$, and as stated in lines 217--224 in the manuscript, we do not propose to alter the model, but retain the un-linearized mapping $f$ to compute the predictive distribution. In other words, **the approximate variational objective does not correspond to a GGN approximation**.
> > >
> > > Note that one of the reasons why it is not desirable to change the model is that if we were to do so, that is, if the model were given by $p(y, \tilde{f}(\cdot)) = p(y | \tilde{f}(X) ) p(\tilde{f}(\cdot)),$ where $\tilde{f}(\cdot) = f(\cdot ; m) + J(\cdot ; m) (\Theta - m)$, then the variational distribution over functions would also have to be defined in terms of $\tilde{f}(\cdot) = f(\cdot ; m) + J(\cdot ; m) (\Theta - m)$ to ensure absolute continuity is preserved. Hence, assuming this model, the only way in which the prior and variational distributions over functions could differ would be in terms of the distributions over $\Theta$ in $\tilde{f}$ while being defined in terms of the same (possibly learnable) parameters $m$ to ensure absolute continuity is preserved, significantly reducing the extent to which the prior and variational distributions over function can differ from one another.
> > >
> > > In contrast, this constraint does not appear in our formulation of the approximation, since we do not change/approximate the model, but instead only approximate the induced distributions over functions in the KL divergence computation.
> > >
> > > ### 2. What's the difference between the proposed method and Immer et al. (2021)?
> > >
> > > Immer et al. (2021) [3] build on [4] in also using the Laplace-GGN approximation. They use the approximate posterior distribution proposed in prior work (e.g., [4]), and their main contribution is to propose to use the linearized (instead of the original) model to construct the posterior predictive distribution. (They do not propose the GGN distribution, nor do they propose a new posterior.) Their main (quite neat) point is that since the GGN approximation can be viewed as exact inference in a different model (which was previously noted in [4]), we may want to use this different model (namely a certain linear model) to construct the predictive distribution.
> > >
> > > **This approach is distinctly different from the approach proposed in our manuscript**: 1. We don't propose to change the model or use a GGN approximation; 2. we obtain a posterior approximation variationally, not using a Laplace approximation; and 3. the linearized mapping is computed at different evaluation parameters each gradient step (and the evaluation parameters are differentiable), whereas Immer et al. (2021) pick a single evaluation point (the MAP solution) and do not update it. The only similarity between Immer et al. (2021) and our method is that a first-order Taylor approximation shows up somewhere along the line, but the way it is used is very distinct.
> > >
> > > ### 3. Significance
> > >
> > > The use of (evolving!) function linearization to approximate implicit distributions is a simple and effective solution to a problem that has long hampered progress in function-space variational inference.
> > >
> > > If the clarifications above, our original response, and the additional experiments on the impact of different context distributions shared above have satisfactorily addressed your questions and concerns, we would be grateful if you would consider updating your score. We would be more than happy to provide any additional information or experiments.
> > >
> > > Thank you for your feedback on our manuscript!
> > >
> > > ### References
> > >
> > > [1] Nocedal, J., Wright, S. J. Numerical optimization, 2006.
> > >
> > > [2] Martens, J. New Insights and Perspectives on the Natural Gradient Method, 2020.
> > >
> > > [3] Immer A., Korzepa, M., Bauer, M. Improving Predictions of Bayesian NNs via Local Linearization, 2021.
> > >
> > > [4] Khan, M. Immer, A., Abedi, E., Korzepa, M. Approximate Inference Turns Deep Networks into GPs, 2020.

---

### Official Review · Reviewer_UrsU · 2022-07-12

**Rating:** 4
**Confidence:** 2
**Soundness:** 2 fair
**Presentation:** 3 good
**Contribution:** 2 fair

**Summary:**

This paper proposes a tractable method for function space Variational inference that allows the method to be applied to realistic real-world settings. The authors achieve this by means of two approximations. Firstly, rather than calculating a KL divergence between two posteriors for which we do not know the closed-form density, they calculate the KL divergence between locally accurate linear approximations to the posteriors, whose density *is* known. Secondly, they use a rough approximation to the supremum over the aforementioned KL divergence, which they argue is enough since it encourages the variational distribution to match the prior on a set of context points.

**Questions:**

## Major

0. [Significance] To me it seems very strange to do function space inference when the function space prior does not have any correlation between inputs. Under this modelling assumption, we are saying that we do not expect one input to tell us anything about the other input. This issue is sidestepped in FSVI due to the fact that NNs have some smoothness properties and as such the approximate posterior does not treat all inputs as independent. Nevertheless, one should strive to have a correct model and then apply approximations which are hopefully not harmful, rather than have an incorrect model that is fixed by applying approximations.

1. [Experimental] It would be great to have a qualitative comparison between FSVI and HMC for some of the to datasets, for example those in Figure 1. Providing only the fits for FSVI without a comparison isn't particularly illustrative, and since these problems are small, HMC can be run to get a (somewhat) good idea of the 'true' posterior.

2. [Experimental] On a similar note, comparisons to Sun, et al. and Ma, et al. on some smaller settings (perhaps the UCI datasets used by Sun et al.). This would allow the reader to understand what is being gained/lost (other than improved tractability) when using FSVI rather than these previous approaches to function space BNNs.

3. [Signifigance] On lines 168-170, the authors suggest that the choice of context distribution can be viewed as a problem-specific modelling choice. I think that this is a very interesting idea, however, the idea is somewhat vague. Would it be possible to elaborate on this and give a few concrete examples?

4. [Experimental/Significance] On lines 170-171, the authors mention that $S$ and $K$ are hyperparameters to be optimised on a validation set. How impactful are these hyperparameters? Do the authors have any concrete pieces of advice for choosing values/ranges? A sensitivity analysis would be very helpful.

5. [Experimental] The ablation study for the choice of context distribution, while a great inclusion, was somewhat unhelpful because there was no accompanying discussion and the plot labels were somewhat mysterious. It would also be interesting to see the impact of swapping the context set and OOD dataset in the OOD results from Tables 1 and 2.

6. [Experimental/Significance] How restrictive is the mean-field variational assumption (c.f., line 174)? Would it be possible to include a small ablation study?

7. [Experimental/Significance] Ditto for the diagonalization discussed on lines 179-180.

8. [Experimental]. For all of the baselines used in Section 5, a description of what hyperparameters were optimised and how (and additionally how hyperparameters which were not optimised were chosen) needs to be supplied. Without this information, it is not clear what to make of the results.

9. [Experimental] Re Figure 2. Why is the brier score being used to measure predictive accuracy, rather than accuracy itself? Also, it seems like FSVI is underfitting, since the accuracy is low for small rotations. However, this seems to contradict the results in Table 1. Is there a difference in the experimental setup that I do not appreciate? Finally, would it be possible to see a plot like this for corrupted CIFAR10? This would be much more informative than a single column in Table 2.

10. [Experimental] Figure 4 is somewhat incomprehensible because there are too many lines. I would suggest removing the ensemble results (other than deep ensemble) since they are not important for the key message.

11. [Experimental/Significance] It seems to me that the conclusion that the correlation between OOD-AUROC and ECE is much less strong for CIFAR10 than FashionMNIST, which calls into question the generality of the finding. Is it possible to run this experiment for additional datasets?

## Minor

A. On line 27, I would suggest citing MacKay in addition to, or instead of, Neal for BNNs.

B. On lines 29-31, the authors suggest that BNNs underperform non-Bayesian methods. However, this is a very strong statement. For example, the Linearised Laplace approximation is known to outperform deep ensembles in some settings; see https://arxiv.org/pdf/2106.14806.pdf and http://proceedings.mlr.press/v139/daxberger21a/daxberger21a.pdf. Similarly, HMC BNNs can also outperform deep ensembles in certain settings http://proceedings.mlr.press/v139/izmailov21a/izmailov21a.pdf.

C. On line 87, the prior is introduced as though it was present in the equation above, which is confusing. Perhaps the authors meant to describe the posterior here?

D. In eqn 1, the authors switch to a KL between the approximate posterior and *prior* without any explanation, which could confuse readers.

E. Proposition 1 does not need to be a proposition. As far as I understand, it is a well-known result (e.g., in the linearised Laplace approximation community). Similarly, Corollary 1 does not need to be a corollary. This is also a well-known result, as evidenced by the trivial proof. In my opinion, presenting these results in such a way only serves to make the paper less approachable.

F. I found the use of $g$ and $\tilde{g}$ in Corollary 1 confusing – the symbols are too similar!

G. Lines 188-189, 198-199 are repeats of 171-172 and can be removed.

H. In what sense is CIFAR100 diverse?  (line 195)

I. Would it be possible to provide a more detailed description of the procedure for generating the monochrome images? I am afraid that the description in the text was not clear to me at all.

J. On line 2017, I believe that there is a typo. Should 'S*M*' be 'S*K*'?

K. It would be great to have a slightly expanded discussion of the related work. In particular, the relation of FSVI to the approaches of, (Benjamin et al., 2019; Pan et al., 2020; Titsias et al., 2020) could be expanded upon.

L. Similarly, a discussion of how the results from (Foong et al., 2019) for weight space BNNs related to function space BNNs, would be helpful.

M. As far as I understand the debate about the properties of mean-field variational distributions in deep BNNs has not been settled, so the statement on line 233 implying otherwise is somewhat strong. I would rather say that "Farquhar et al., (2020) suggest that deep BNNs do not suffer from this issue."

N. On lines 246-247, do the authors mean that the *uncertainty* of the predictions will be significantly higher?

O. Lines 258-260, seem to be redundant with the previous paragraph.

P. I assume that the Kaggle dataset on line 316 is the EyePACS dataset on line 313?


**Limitations:**

The authors have not made an effort to explicitly describe the limitations of their work. For example, they have not discussed the sensitivity of their proposed method to its hyperparameters, nor have they described the trade-offs between their method and existing functional BNN methods. However, I am confident that this issue can easily be addressed.

**Strengths And Weaknesses:**

## Strengths

**Originality**: the paper proposes a novel solution to a challenging and important problem in the BNN literature. I particularly enjoyed the idea of using local linearization to approximate the KL divergence between the approximate posterior and prior!

**Clarity**: for the most part I found the paper well written and explained. There were a few issues, which I detail later.

## Weaknesses

**Quality**: While the experimental evaluation is quite extensive (many benchmarks and many baselines – very nice!), I do have some concerns and suggestions, which are also listed below. My main concern is that the experiments do not add to the understanding of the performance of the method–the relative strengths and weaknesses are not apparent, and the impact of the various approximations is unclear. There is also a lack of some important details that make it difficult to interpret the results of the experiments.

**Significance**: I have a number of concerns about the approximations and assumptions in this work, which have not been addressed in the text. Without having these concerns addressed it is hard for me to make any judgement about the potential significance of the work. My primary concern is with the choice of prior. This concern and others are listed in the questions below.

---

> ### Author Response · Authors · 2022-08-03
> **Response to Reviewer UrsU (5/5)**
>
> ### 12. Correlation between ECE and OOD Detection Accuracy
>
> **Quote from review:**
> > [Experimental/Significance] It seems to me that the conclusion that the correlation between OOD-AUROC and ECE is much less strong for CIFAR10 than FashionMNIST, which calls into question the generality of the finding. Is it possible to run this experiment for additional datasets?
>
> As requested, we ran the experiment for a model trained on MNIST (instead of FashionMNIST) via FSVI using the same context distribution (“random monochrome”, i.e.,images generated from the training data pixel distribution).
>
> The results of our experiments can be found here: [fully-anonymized link to interactive results dashboard (Correlation)](https://rebrand.ly/sensitivity_analysis_correlation).
>
> Similar to the trend observed when training on FashionMNIST, both test ECE and NLL increase for smaller prior variance parameters. In contrast to the models trained on FashionMNIST with the "random monochrome" context distribution, we observe a slight decrease in OOD detection accuracy for the largest prior variance parameters on FashionMNIST and KMNIST.
>
> We suspect that the trend for CIFAR-10 may be substantially less clear, since out-of-distribution detection accuracy varies much more heavily from epoch to epoch (and even from gradient step to gradient step) than when training on FashionMNIST and MNIST and presenting aggregates over multiple seeds may help bring out a trend more clearly.
>
>
> ### 13. General note on experiments
>
> We hope to rerun all of our experiments over the next couple months (subject to compute constraints)  to ensure there are no inconsistencies between results from a few months ago and results obtained now. We would appreciate any additional suggestions for experiments to include.
>
> ### 14. Minor comments
>
> We agree with most comments and will make the requested changes.
>
> A few additional comments:
>
> **Quote from review:**
> > H. In what sense is CIFAR100 diverse? (line 195)
>
> By diverse, we mean that the images show a relatively diverse set of depictions, ranging from outdoor scenes, to animals, insects, and vegetables.
>
>
> **Quote from review:**
> > I. Would it be possible to provide a more detailed description of the procedure for generating the monochrome images? I am afraid that the description in the text was not clear to me at all.
>
> To generate a sample from the “monochrome images” distribution, we first take all images in the training data, flatten each image, and stack the flattened images into a single vector. We then draw a random element (i.e., a pixel) from this vector and then use this pixel to generate a monochrome image of a given resolution by setting every pixel of the image equal to the value of the pixel that was drawn.
>
>
> ### 15. Limitations
>
> **Quote from review:**
> > The authors have not made an effort to explicitly describe the limitations of their work. For example, they have not discussed the sensitivity of their proposed method to its hyperparameters, nor have they described the trade-offs between their method and existing functional BNN methods. However, I am confident that this issue can easily be addressed.
>
> While a direct comparison to other function-space variational methods is difficult, we hope that our empirical comparison to SPG (Ma and Hernández-Lobato, 2021) in Tables 1 and 2 and to Sun et al. (2019) on regression problems further elucidates the differences between our approximation and alternative function-space variational inference methods.
>
> Since paper revisions are still limited to 9 pages, including the additional results and discussions in the paper without a major rewriting of the draft is difficult, but we commit to including the extended discussions and new results in the updated version.
>
> We hope that the responses to your questions above and our discussion of the requested results (such as the effect of the choice of context distribution and other hyperparameters on performance) were able to address your concerns. If not, we would appreciate any further pointers as to how to further improve our discussion of limitations.

---

> ### Author Response · Authors · 2022-08-03
> **Response to Reviewer UrsU (4/5)**
>
> ### 8. Diagonalization assumption
>
> **Quote from review:**
> > [Experimental/Significance] Ditto for the diagonalization discussed on lines 179-180.
>
> Since submitting the paper, we have been able to improve the efficiency of the covariance computation, which is the main bottleneck in the approximation. The diagonal approximation reduces the computational cost, since it only requires computing the diagonal of the covariance matrix. With the lowered computational cost, we are now able to compute the KL efficiently using the full covariance of the variational distribution for small to moderately-sized sets of context points, depending on the number of neural network parameters.
>
> To investigate whether using the full covariance provide any empirical benefits, we compared the predictive performance (in terms of predictive accuracy, expected calibration error, and out-of-distribution uncertainty quantification) between models trained via FSVI with and without using a diagonal approximation.
>
> The results of our experiments can be found here: [fully-anonymized link to interactive results dashboard (Full Covariance)](https://rebrand.ly/sensitivity_analysis_context_distribution_full_cov).
>
> For this experiment, we trained a model with FSVI on CIFAR-10 using a ResNet-19 architecture and used the “Miniplaces” dataset as a context distribution. All hyperparameters in the two sets of experiments (i.e., with and without using a diagonal approximation) are the same.
>
> As can be seen in the figures, using the full covariance in this setting leads to an improvement in performance, especially in terms of test accuracy, but also in terms of uncertainty quantification.
>
> ### 9. Range of Hyperparameters
>
> **Quote from review:**
> > [Experimental]. For all of the baselines used in Section 5, a description of what hyperparameters were optimised and how (and additionally how hyperparameters which were not optimised were chosen) needs to be supplied. Without this information, it is not clear what to make of the results.
>
> For DUQ, SPG, VOGN, BNN-Laplace, we report the results from the respective papers. For MFVI, MC Dropout, MAP, SWAG, and Radial BNNs, we followed the same validation protocol as for FSVI (see Appendix F.1.), but we did not keep a record of the hyperparameter ranges used. We will endeavor to reproduce the results with another round of hyperparameter tuning and report the ranges used. Broadly speaking, in addition to optimizing general hyperparameters, such as the learning rate, for MFVI, we optimized the prior variance and the KL tempering hyperparameter, for Radial BNNS, we optimized the prior variance, for for MC Dropout, we optimized the dropout probability, and for MAP and SWAG, we optimized the $L_2$ regularization coefficient.
>
> ### 10. Rotated MNIST Experiment
>
> **Quote from review:**
> > [Experimental] Re Figure 2. Why is the brier score being used to measure predictive accuracy, rather than accuracy itself?
>
> We used the Brier score, since it is more sensitive to changes in the predictions than predictive accuracy, and the difference in performance between methods was difficult to distinguish visually.
>
> **Quote from review:**
> > Also, it seems like FSVI is underfitting, since the accuracy is low for small rotations. However, this seems to contradict the results in Table 1. Is there a difference in the experimental setup that I do not appreciate? Finally, would it be possible to see a plot like this for corrupted CIFAR10? This would be much more informative than a single column in Table 2.
>
> Table 1 shows results for models trained on FashionMNIST, whereas Figure 2 shows results for models trained on MNIST.
>
> We have not yet been able to produce the corrupted CIFAR-10 results, but we hope to include it in the revised draft. We provide the accuracy under the second-highest corruption of CIFAR-10 in Table 2 (last column).
>
>
> ### 11. RETINA results
>
> **Quote from review:**
> > [Experimental] Figure 4 is somewhat incomprehensible because there are too many lines. I would suggest removing the ensemble results (other than deep ensemble) since they are not important for the key message.
>
> We appreciate the feedback and will make the change as requested. For reference, we also provide complementary tabular results in Appendix B.1.

---

> ### Author Response · Authors · 2022-08-03
> **Response to Reviewer UrsU (3/5)**
>
> ### 5. Sensitivity analysis of supremum estimation
>
> **Quote from review:**
> > [Experimental/Significance] On lines 170-171, the authors mention that S and K are hyperparameters to be optimised on a validation set. How impactful are these hyperparameters? Do the authors have any concrete pieces of advice for choosing values/ranges? A sensitivity analysis would be very helpful.
>
> We performed an evaluation of the effect of different choices of S and K on uncertainty quantification in appendices D.4. and D.5. We also performed an additional set of new experiments, which can be found here: [fully-anonymized link to interactive results dashboard](https://rebrand.ly/sensitivity_analysis_supremum_estimation).
>
> While this analysis is not exhaustive, it is consistent with the behavior we observed in previous experiments on several different datasets:
>
> 1. There does not appear to be a consistent change in the quality of uncertainty quantification (as measured in ECE and OOD detection accuracy) as the maximum is computed over a larger collection of sets of context points (i.e., larger $S$).
> 2. There is an increase in ECE as the number of context points is increased (i.e., larger $K$). This is potentially due to the diagonal approximation used here, as the KL divergence between Gaussian distributions with diagonal covariance grows with the number of evaluation points.
>
> We will include these results as polished figures in the revised paper.
>
> ### 6. Ablation study on the effect of the context distribution
>
> **Quote from review:**
> > [Experimental] The ablation study for the choice of context distribution, while a great inclusion, was somewhat unhelpful because there was no accompanying discussion and the plot labels were somewhat mysterious. It would also be interesting to see the impact of swapping the context set and OOD dataset in the OOD results from Tables 1 and 2.
>
> Thank you for pointing this out, we will update the labels and include a discussion.
>
> 1. The label “train-pixel” refers to a context distribution constructed from the union of the training data and a dataset of monochrome images whose color is sampled from the pixel distribution of the training data.
> 2. The label “train-uniform” refers to a context distribution constructed from the union of the training distribution and randomly generated images whose pixels are drawn i.i.d. from a uniform distribution between -1 and 1.
> 3. The label “train-ood” refers to a context distribution constructed from the union of the training distribution and some out-of-distribution data (e.g., KMNIST or CIFAR-100, as in Tables 1 and 2).
>
> We complemented the results in the paper with additional experiments, as requested. While we cannot simply swap the context distributions used in Tables 1 and 2, since KMNIST contains grayscale and CIFAR-100 RBG images, we instead replaced KMNIST with MNIST, NotMNIST, and FashionMNIST (i.e., the training data), respectively, and replaced CIFAR-100 with downscaled images from the “Miniplaces” dataset, which contains a large number of images of places and objects.
>
> The results of our experiments can be found here:
> 1. [fully-anonymized link to interactive results dashboard (FashionMNIST)](https://rebrand.ly/sensitivity_analysis_context_distribution_fmnist).
> 2. [fully-anonymized link to interactive results dashboard (CIFAR-10)](https://rebrand.ly/sensitivity_analysis_context_distribution_full_cov).
>
> The results show that the reliability of out-of-distribution uncertainty varies heavily across context distributions and that the “Miniplaces” context distribution achieves a similar performance (though slightly worse) as when using CIFAR-100.
>
> ### 7. Impact of mean-field assumption
>
> **Quote from review:**
> > [Experimental/Significance] How restrictive is the mean-field variational assumption (c.f., line 174)? Would it be possible to include a small ablation study?
>
> Unfortunately, our codebase makes it challenging to implement non-mean-field variational distributions and we were unable to conduct this experiment.
>
> However, Farquhar et al. (2020) showed that a mean-field distribution over network parameters can result in a distribution over functions with covariance structure and our 1D experiments suggest that the poor performance of parameter-space mean-field variational inference may not be due to the variational assumption but due to the objective.

---

> ### Author Response · Authors · 2022-08-03
> **Response to Reviewer UrsU (2/5)**
>
> ### 2. Comparison to HMC
>
> **Quote from review:**
> > [Experimental] It would be great to have a qualitative comparison between FSVI and HMC for some of the to datasets, for example those in Figure 1. Providing only the fits for FSVI without a comparison isn't particularly illustrative, and since these problems are small, HMC can be run to get a (somewhat) good idea of the 'true' posterior.
>
> We appreciate the suggestion and compared both the posterior predictive distribution of function-space variational inference with a mean-field Gaussian variational distribution and parameter-space variational inference with a mean-field Gaussian variational distribution to the HMC posterior predictive distribution. All methods use a Gaussian prior with zero mean and a variance of 10.
>
> The results of our experiments can be found here:
> 1. [fully-anonymized link to Snelson 1D regression with FSVI](https://drive.google.com/file/d/1DfCUcg-3S2nsv_TuqyZ_P3-TpCI8itek/view?usp=sharing).
> 2. [fully-anonymized link to Snelson 1D regression with PSVI](https://drive.google.com/file/d/1BZSGZk2sD_7UsGf5Ki4DHuVuLxZmpRG9/view?usp=sharing).
> 3. [fully-anonymized link to Snelson 1D regression with HMC](https://drive.google.com/file/d/1k_geB1tmNGm4R6yXXREi8AiCmxcfXjbb/view?usp=sharing).
> 4. [fully-anonymized link to Oat 1D regression with FSVI](https://drive.google.com/file/d/1a-VR5seyphjaJJfco-kDcpoMfZhhhDu5/view?usp=sharing).
> 5. [fully-anonymized link to Oat 1D regression with PSVI](https://drive.google.com/file/d/1EAi2gdYL-pFoVvPE6mpwgb6TQwWyvlcO/view?usp=sharing).
> 6. [fully-anonymized link to Oat 1D regression with HMC](https://drive.google.com/file/d/1z7U1kgp1sa6xLa6087xiUtwj3qKc3MmT/view?usp=sharing).
>
> The results show that FSVI posterior predictive distribution (which was trained without making a diagonal approximation to compute the KL divergence) is somewhat close to the HMC posterior, whereas the PSVI posterior predictive distribution is not. Notably, while the FSVI and HMC posterior predictive distributions look somewhat similar, the HMC predictive distribution appears to fit the training data somewhat less well.
>
> We will include these results as polished figures in the revised paper.
>
>
> ### 3. Comparison to Sun et al. (2019)
>
> **Quote from review:**
> > [Experimental] On a similar note, comparisons to Sun, et al. and Ma, et al. on some smaller settings (perhaps the UCI datasets used by Sun et al.). This would allow the reader to understand what is being gained/lost (other than improved tractability) when using FSVI rather than these previous approaches to function space BNNs.
>
> We performed the requested comparison to Sun et al. (2019). Please see our comment to all reviewers for our results. We included results for several different hyperparameter configurations to elucidate the sensitivity of the predictive performance under FSVI under different hyperparameters.
>
> We will include these results in the revised paper.
>
>
> ### 4. Context distribution as a modeling choice
>
> **Quote from review:**
> > [Signifigance] On lines 168-170, the authors suggest that the choice of context distribution can be viewed as a problem-specific modelling choice. I think that this is a very interesting idea, however, the idea is somewhat vague. Would it be possible to elaborate on this and give a few concrete examples?
>
> In real-world settings, data scientists and machine learning engineers typically have deep knowledge of the task at hand, and there are a plethora of applications, where only a small labeled dataset but large amounts of related unlabeled data is available. We believe that the choice of context distribution can be informed by the problem at hand.
>
> For example, if we wish to design a model that is able to correctly diagnose diabetic retinopathy, a medical condition that can lead to blindness, from patients’ retina scans, we might want to use a model trained on data gathered in one location with one specific type of measurement equipment in a different location where the goal is to classify input samples (i.e., retina scans), obtained using a different type of measurement equipment. In such a scenario, we may be able to get access to unlabeled data from the test location and retrain the model using the old training data along with the new, unlabeled data, which could be used to construct the context distribution.

---

> ### Author Response · Authors · 2022-08-03
> **Response to Reviewer UrsU (1/5)**
>
> ### General Comments
>
> Thank you for your very careful and detailed review as well as for your questions and suggestions.
>
> We were pleased to see that you found that our paper “proposes a novel solution to a challenging and important problem in the BNN literature” and that you “particularly enjoyed the idea of using local linearization to approximate the KL divergence between the approximate posterior and prior.” This is much appreciated.
>
> We conducted a significant number of additional experiments in an effort to, hopefully, satisfactorily answer your questions. We hope our discussion and clarifications below address your questions and concerns and would be more than happy to answer any follow-up questions you may have.
>
>
> ### 1. Prior variance
>
> **Quote from review:**
> > [Significance] To me it seems very strange to do function space inference when the function space prior does not have any correlation between inputs. Under this modelling assumption, we are saying that we do not expect one input to tell us anything about the other input. This issue is sidestepped in FSVI due to the fact that NNs have some smoothness properties and as such the approximate posterior does not treat all inputs as independent. Nevertheless, one should strive to have a correct model and then apply approximations which are hopefully not harmful, rather than have an incorrect model that is fixed by applying approximations.
>
> This is a good point, and we agree with your statement that “one should strive to have a correct model and then apply approximations which are hopefully not harmful, rather than have an incorrect model that is fixed by applying approximations”
>
> We want to clarify that at no point do we propose to change the model; we merely approximate the distribution over functions under a given mapping and prior distribution over parameters by the distribution induced by the same distribution over parameters under a linearization of the mapping. More specifically, we do not **posit** that the distribution over linearized mappings is given by a distribution over functions with diagonal covariance functions—this is the actual distribution over linearized functions induced by prior means with zero mean parameters.
>
> This is because the Jacobian of a neural network with a final linear layer with bias evaluated at a zero prior mean, will be zero for all network parameters except the final-layer bias parameters, where the entries of the Jacobian will be unity. Therefore, when computing $J \Sigma J^\top$, we will get an n-by-n (for n evaluation points) covariance matrix, which, for output dimension $j$ is zero everywhere except on the diagonal, where (since $1 \times \Sigma_{j} \times 1$), the matrix will be equal to $\Sigma_{j}$. (We have uploaded a revised version of the appendix that derives the covariance matrix in this setting in full detail.) On a related note, we want to emphasize that only the distribution over **linearized** functions will have a diagonal covariance, but the distribution over non-linearized functions under even a zero-mean prior distribution (i.e., the intractable model) will not.
>
> All this being said, we would agree that it would likely be preferable to be able to have a prior distribution over linearized mapping that conveys non-zero covariance information and also explored prior distributions over parameters with non-zero means, so that we can obtain an induced distribution over (linearized) functions with a non-diagonal covariance function.
>
>  However, except for the continual learning task considered in the paper, we have not been able to outperform the zero-mean prior over the network parameters in our experiments. In particular, we found that induced prior distributions over functions (induced by a prior distribution over parameters with non-zero mean) that were most similar to the diagonal distributions worked best. For example, in our experiments, we found little difference in terms of predictive performance between distributions over functions induced by a distribution over parameters with very small, non-zero mean and a variance that would induce a diagonally-dominated covariance function and a distribution over functions with a diagonal covariance whose variance is in the same order of magnitude as the diagonal of the induced non-diagonal induced covariance function.

---

> > ### Comment · Reviewer_UrsU · 2022-08-08
> > **Further discussion (1/5)**
> >
> > Thank you for the detailed response and clarifications.
> >
> > > However, except for the continual learning task considered in the paper, we have not been able to outperform the zero-mean prior over the network parameters in our experiments. In particular, we found that induced prior distributions over functions (induced by a prior distribution over parameters with non-zero mean) that were most similar to the diagonal distributions worked best. For example, in our experiments, we found little difference in terms of predictive performance between distributions over functions induced by a distribution over parameters with very small, non-zero mean and a variance that would induce a diagonally-dominated covariance function and a distribution over functions with a diagonal covariance whose variance is in the same order of magnitude as the diagonal of the induced non-diagonal induced covariance function.
> >
> > This is very surprising to me, but I am also not 100% sure I understand what you are saying. Could you please elaborate on the specifics of the experiments that you ran here, and perhaps also provide some of your own intuition for the results?

---

> > > ### Author Response · Authors · 2022-08-08
> > > **Response re: Prior Variance**
> > >
> > > Thank you for engaging with our reply! We'll gladly elaborate further.
> > >
> > > In the comment above, we were referring to experiments conducted to elucidate whether using a prior distribution with a full covariance may result in an improvement in predictive performance and/or uncertainty quantification.
> > >
> > > Specifically, we explored three different ways of computing the KL divergence (in order of least to most computationally expensive):
> > >
> > > 1. Using a **zero-mean** prior distribution over parameters (which, as explained above, results in a distribution over linearized functions with a diagonal covariance) and applying a diagonal approximation to both the prior and variational covariance over linearized functions (the former of which is already diagonal);
> > >
> > > 2. Using a **zero-mean** prior distribution over parameters (which, as explained above, results in a distribution over linearized functions with a diagonal covariance) and **not** applying a diagonal approximation.
> > >
> > > 3. Using a **non-zero-mean** prior distribution over parameters and **not** applying a diagonal approximation.
> > >
> > > In our response below, we provided experiments demonstrating the difference between 1. and 2. (see [[link to response]](https://openreview.net/forum?id=OQs0pLKGGpS&noteId=TrgkmymYmgi)).
> > >
> > > In the quoted comment above, we were referring to experiments comparing options 1. and 3. In particular, since a prior distribution over parameters with a non-zero mean is needed to obtain a prior distribution over linearized functions with a non-diagonal covariance, we chose the prior distribution mean parameters to be small values (on the order of 0.001-0.01) that ensure that the prior predictive mean remains close to zero. We then varied the variance of the prior distribution over parameters to see which parameter variance would lead to the best predictive performance and uncertainty quantification. Although the best parameter variance varied between neural network architectures, we found that the variance parameters that tended to work best tended to be the ones that would induce a prior predictive variance that is of the same order of magnitude as the predictive variance that worked best under options 1. and 2. Additionally, we found that prior predictive covariances that were more diagonally dominated (i.e., more similar to the prior predictive covariance under option 2.) tended to work better than less diagonally-dominated covariances.
> > >
> > > We, too, were initially surprised that there did not appear t be a noticeable benefit from option 3., but we believe that there may be two (intuitive) reasons for this:
> > >
> > > 1. The neural network architectures used for the classification tasks considered in the paper (small CNNs and ResNet-18) have strong inductive biases (or "smoothness properties", as you noted, depending on the network architecture) and regardless of the KL approximation and prior distribution chosen, the neural network architecture constrains the variational distribution over functions and induces a certain covariance structure the BNN has to adhere to. Since the prior and variational distributions over functions need to be defined in terms of the same mapping $f$ (i.e., the mapping defined by the neural network architecture) and the variational distribution also appears in the expected log-likelihood term of the variational objective, the prior predictive distribution's off-diagonal covariance entries may have a small effect on training in practice.
> > >
> > > 2. For the classification problems considered in the paper, the KL evaluation points sampled from the sets of context points can be "far away" from one another (in image space). As a result, the predictive covariance between them tends to be small in magnitude compared to their variance, resulting in a diagonally-dominated covariance matrix.
> > >
> > > We hope this response answers your question. Please let us know if anything remains unclear.

---

### Official Review · Reviewer_uYCu · 2022-07-13

**Rating:** 6
**Confidence:** 3
**Soundness:** 3 good
**Presentation:** 3 good
**Contribution:** 3 good

**Summary:**

The manuscript proposes a scalable functional-space variational inference method by incorporating prior information. The main contribution of the paper is to propose a KL divergence estimator between a posterior and variational distribution over functions by approximating the first-order Tayler series expansion of the mean of the parameters.


**Questions:**

What are the potential ways to apply this approach to non-Gaussian priors?
How does the proposed method work for other problems such as Bandits, and Bayesian Optimization?

**Ethics Review Area:**

["I don’t know"]

**Limitations:**

The results presented are convincing for the image classification task, but you need to compare against other variational inference approaches proposed in the literature. Some references are provided above.



**Strengths And Weaknesses:**

The proposed approach seems intuitive and the proof seems reasonable. The results demonstrate the benefits of the approach.

Weakness:
-  The diagonalization approximation to the covariance matrices in Section 3.2 is a simplification that deviates from the original approach from Sun et. al.
- The proposed method are presented only image classification problems.
-  Authors need to compare other approaches that scale functional variation inference to even larger models: please refer to the following manuscript.
"Carvalho, Eduardo DC, et al. "Scalable uncertainty for computer vision with functional variational inference." CVPR'2020

- Other approaches which are not functional variational inference but provide well-calibrated uncertainty estimation need to be compared.

Karandikar, Archit, et al. "Soft calibration objectives for neural networks." Advances in Neural Information Processing Systems 34 (2021): 29768-29779.
Krishnan, Ranganath, and Omesh Tickoo. "Improving model calibration with accuracy versus uncertainty optimization." Advances in Neural Information Processing Systems 33 (2020): 18237-18248.

---

> ### Author Response · Authors · 2022-08-02
> **Response to Reviewer uYCu (2/2)**
>
> ### 4. Non-Gaussian distributions over parameters
>
> **Quote from review:**
> > What are the potential ways to apply this approach to non-Gaussian priors?
>
> The result in Proposition 1 is applicable to any member of the location-scale family of distributions, since any such distribution is preserved under affine transformations.
>
> Therefore, examples of other prior and variational distributions over parameters that are admissible under the approximation proposed in the paper are Laplace and Cauchy distributions. The only requirement is that the prior and variational distributions over parameters be absolutely continuous to ensure the function-space KL is well-defined.
>
> One drawback of using alternative distributions over parameters, such as a cauchy or Laplace distribution, is that, if we don’t wish to make a diagonal approximation to the covariance of the induced distribution over functions, we would have to compute a KL divergence between multivariate random variables distributed according to multivariate Cauchy or Laplace distributions. Unfortunately, we do not have exact analytical expressions for the KL divergence between two multivariate Cauchy or two multivariate Laplace distributions, meaning that we would have to approximate the KL, for example by using Monte Carlo estimation.
>
> ### 5. Extension to other settings
>
> **Quote from review:**
> > How does the proposed method work for other problems such as Bandits, and Bayesian Optimization?
>
> This is a good question. While FSVI can be used in these settings the same way as any other Bayesian neural network would be used, FSVI provides the added benefit of allowing the careful design of a context distribution. For example, if we have any prior information about the state space in a bandit setting, we can use FSVI to train a good policy and use the knowledge about the state space to design a context distribution that systematically covers the most relevant parts of the state space. If even more prior information is available, for example, in the shape of expert demonstrations in a bandit or RL setting, such expert demonstrations can be distilled into a function that can be used as an empirical prior policy. Bayesian optimization might benefit in similar ways if prior information is available.

---

> ### Author Response · Authors · 2022-08-02
> **Response to Reviewer uYCu (1/2)**
>
> ### General Comments
>
> Thank you for your thoughtful and positive review and for your questions. We were pleased you found the proposed method “intuitive” and that our “results demonstrate the benefits of the approach.”
>
> We hope our clarifications below address your questions and any remaining concerns and would be more than happy to answer any follow-up questions you may have.
>
> ### 1. Diagonal approximation
>
> **Quote from review:**
> > The diagonalization approximation to the covariance matrices in Section 3.2 is a simplification that deviates from the original approach from Sun et. al.
>
> Since submitting the paper, we have been able to improve the efficiency of the covariance computation, which is the main bottleneck in the approximation. The diagonal approximation reduces the computational cost, since it only requires computing the diagonal of the covariance matrix. With the lowered computational cost, we are now able to compute the KL efficiently using the full covariance of the variational distribution for small to moderately-sized sets of context points, depending on the number of neural network parameters.
>
> To investigate whether using the full covariance provide any empirical benefits, we compared the predictive performance (in terms of predictive accuracy, expected calibration error, and out-of-distribution uncertainty quantification) between models trained via FSVI with and without using a diagonal approximation.
>
> The results of our experiments can be found here: [fully-anonymized link to interactive results dashboard](https://rebrand.ly/sensitivity_analysis_context_distribution_full_cov).
>
> For this experiment, we trained a model with FSVI on CIFAR-10 using a ResNet-19 architecture and used the “Miniplaces” dataset as a context distribution. All hyperparameters in the two sets of experiments (i.e., with and without using a diagonal approximation) are the same.
>
> As can be seen in the figures, using the full covariance in this setting leads to a notable improvement in performance, especially in terms of test accuracy, but also in terms of uncertainty quantification.
>
> ### 2. Regression problems
>
> **Quote from review:**
> > ​​The proposed method are presented only image classification problems.
>
> We’ve taken your feedback on board and have conducted several regression experiments and compared the results using our approximation with that of Sun et al. (2019). We found that our method outperforms the approximation proposed by Sun et al. (2019) on 5 out 6 datasets considered. Please see our comment to all reviewers for our results. We aim to perform additional regression experiments and include them in the paper.
>
> ### 3. Comparison to additional methods
>
> **Quote from review:**
> > Authors need to compare other approaches that scale functional variation inference to even larger models: please refer to the following manuscript. "Carvalho, Eduardo DC, et al. "Scalable uncertainty for computer vision with functional variational inference." CVPR'2020
>
> There may be a misunderstanding. The ResNet-18 models used in our CIFAR-10 experiments are larger than the models used in the reference above. Including mean and variance parameters, the ResNet18 models trained for our experiments have over 20 million parameters. We, of course, compared our method to other models of the same size: All models in Figure 2 (expect for SPG) use a ResNet-18 architecture.
>
> We attempted to run experiments with the code provided by the authors of the paper referenced above, but were unable to get the experiments to work on time. We will endeavor to compare our approximation to the method proposed by the authors and will include the paper in our discussion of related work.
>
> **Quote from review:**
> > Other approaches which are not functional variational inference but provide well-calibrated uncertainty estimation need to be compared.
>
> > The results presented are convincing for the image classification task, but you need to compare against other variational inference approaches proposed in the literature.
>
> There seems to be a misunderstanding. We compared our method to 9 methods (MAP, MFVI, tempered MFVI, MC Dropout, SWAG, VOGN, DUQ, BNN Laplace, and Deep Ensembles) that do not use function-space variational inference (see Tables 1 and 2) and to four other variational inference methods (MFVI, tempered MFVI, MC Dropout, VOGN).

---

> > ### Comment · Reviewer_uYCu · 2022-08-10
> > **thank you for sharing the additional results**
> >
> > Thank you for sharing additional results. I'll leave it to area chairs to decide whether to admit additional results with changes to the formulation from the original submission.
> >
> > Regarding comparison with other methods - is more bring to your attention the approaches which provide well-calibrated uncertainty scores even though they may be using similar algorithms (e.g MFVI).

---

> > > ### Author Response · Authors · 2022-08-10
> > > **Thank you for your reply!**
> > >
> > > **Thank you for replying to our response!**
> > >
> > > ### Clarification
> > >
> > > > I'll leave it to area chairs to decide whether to admit additional results with changes to the formulation from the original submission.
> > >
> > > We would like to emphasize that **we have not changed the formulation of the original submission**. The variant of the KL divergence approximation for which we provided additional results in our response above **complement** the results in the original submission. As per the submission guidelines, the inclusion of additional results is allowed and we intend to include the results in the revised draft.
> > >
> > > > Regarding comparison with other methods - is more bring to your attention the approaches which provide well-calibrated uncertainty scores even though they may be using similar algorithms (e.g MFVI).
> > >
> > > We're not sure what you mean exactly, but we would like to emphasize that we compared our method to several other methods that have ben shown to provide well-calibrated uncertainty scores (e.g., DUQ, SWAG, VOGN, deep ensembles) that use distinctly different algorithms. We endeavor to include additional comparisons, as per your suggestions.
> > >
> > > We also note that we have uploaded a revised version of the manuscript that cites the work by Carvalho et al. (2020) [1].
> > >
> > > ### Any remaining questions or concerns?
> > >
> > > If the additional results you requested (e.g., the comparison to Sun et al. (2019) [2] and the full-covariance experiments) and our clarifications regarding the large set of other methods we compared against, the compatibility of our approach with non-Gaussian distributions over parameters, and extensions to other settings (e.g., bandits and Bayesian optimization) have addressed your questions and concerns, we would be grateful if you would consider updating your score. We would be more than happy to provide any additional information or experiments if any remaining questions or issues remain.
> > >
> > > **Thank you for your feedback on our manuscript and for helping us improve our work!**
> > >
> > > ### References
> > >
> > > [1] Carvalho, E. D. C., Clark R., Nicastro, A.,Kelly, P. H. J. Scalable Uncertainty for Computer Vision with Functional Variational Inference. CVPR, 2020.
> > >
> > > [2] Sun, S., Zhang, G., Shi, J., Grosse, R. Functional Variational Bayesian Neural Networks. ICLR, 2019.

---

### Official Review · Reviewer_UY8r · 2022-07-13

**Rating:** 7
**Confidence:** 3
**Soundness:** 4 excellent
**Presentation:** 4 excellent
**Contribution:** 4 excellent

**Summary:**

The authors propose a change to an established approach (Sun et al., 2019) for performing inference in Bayesian neural networks, which works directly in function-space rather than weight-space. The previous work by Sun et al. (2019) derive an ELBO with an intractable KL divergence between processes, so to make the problem tractable, they estimate the gradients of this KL term using the spectral Stein gradient estimator. This estimator, however, has been shown to be less efficient for high-dimensional distributions.
As an alternative approach, the current paper's authors propose a simple estimator of the KL term, which is based on a Taylor expansion of the random functions induced by BNNs and an MC estimate over context points from the input space.
The authors empirically show how their method leads to state-of-the-art uncertainty estimation and predictive performance on several datasets.


**Questions:**

1. You mention that you choose a prior variance of 10, and based on the experiment in section 5.4 it is clear how this choice is seemingly the best. I am a little confused about whether this is the prior over weights of the BNNs or over functions they represent, however. I suppose it is over the weights, but the text keeps referring to the induced prior over functions. Which prior variance does $\Sigma_0$ refer to?

2. Further to the question above, if $\Sigma_0$ is the variance of the prior over the weights, did you try to estimate the induced variance over functions as well? Given that the datasets are normalised, an induced function-space variance of more than 1 would be peculiar.

3. In lines 154-157, you mention that, theoretically, the approximation to the KL divergence will be better for smaller variances. While a variance of 10 empirically works the best, did you investigate the quality of the approximation for this value? I'm wondering whether the variance of 10 might actually result in a rather poor approximation that happens to work well and whether this is somehow related to the cold posterior effect (this is more of a speculative/discussion point rather than a question and I might well be misunderstanding something here!).

4. Did you empirically compare the performance of your approach to that of Sun et al. (2019) on problems that are within reach of both approaches?


**Limitations:**

The authors have adequately addressed the limitations of their work. A potential negative societal impact has not been discussed, but it is not necessary given the paper's theoretical nature.

**Strengths And Weaknesses:**

**Strengths**

This is an impressive paper; incredibly well-written and thorough, with a good and concise related work section. The proposed approach is a novel and significant contribution to the field, which will undoubtedly be of great interest to the NeurIPS community.

The proposed approach is carefully derived in a way that makes it possible for the reader to follow along all the way. The paper contains a great empirical assessment with an impressive amount of details for making the work reproducible. When a setting or a choice of, say, a hyperparameter or distribution is not immediately clear, the authors carefully discuss the options. This might be the most polished and detailed paper I have ever reviewed.

The proposed approach also works well in practice, and several studies of the effect of hyperparameters and priors (in the supplementary) provide valuable insights.

**Weaknesses**

It's really quite hard to find any substantial weaknesses in the paper.
In my view, the greatest weakness is that the authors do not compare their approach against that of Sun et al. (2019) despite framing the entire paper as an improvement over this. It would have been interesting to see the two approaches compared side-by-side on problems that both methods can handle.

An additional (minor!) weakness of the proposed approach is that it introduces a few extra choices to be made and hyperparameters to be optimised compared to the method by Sun et al. (2019). The authors do, however, provide suggestions and thorough discussions of how to make these choices.

**Comments, typos, and various minor things**

* Line 100: $\mathcal Q_\theta$ -> $\mathcal Q_{q_\Theta}$.
* Line 111: The word "poses" doesn't seem to fit in this sentence.
* In the equation following line 138, the matrix $\mathbf S$ is only defined in the appendix.
* In line 175, if $\Sigma$ is the variance, I think the reparametrisation should be written (informally) using $\Sigma^{½}$ or using the Cholesky decomposition $L$.
* Line 207: $SM$ should be $SK$, I suppose.
* In lines 246-247, I find the phrasing of "predictions will be significantly higher" a bit confusing. I suppose it refers to the predictive uncertainty, which should be higher, but I'm not entirely sure.
* I suppose the ablation studies are actually more like hyperparameter tuning studies rather than actual ablation studies.

---

> ### Author Response · Authors · 2022-08-02
> **Response to Reviewer UY8r (2/2)**
>
> ### 2. Quality of KL approximation
>
> **Quote from review:**
> > In lines 154-157, you mention that, theoretically, the approximation to the KL divergence will be better for smaller variances. While a variance of 10 empirically works the best, did you investigate the quality of the approximation for this value? I'm wondering whether the variance of 10 might actually result in a rather poor approximation that happens to work well and whether this is somehow related to the cold posterior effect (this is more of a speculative/discussion point rather than a question and I might well be misunderstanding something here!).
>
> We empirically tested the accuracy of the linearization and found that---as expected---it is relatively poor for variance values greater than $\approx 1$. However, as described above, whether we use a prior with a diagonally dominated covariance functions induced by a prior distribution over parameters with non-zero mean or a diagonal covariance function induced by a prior distribution over parameters with zero mean that has a similar variance magnitude makes little difference in practice.
>
> However, what does make a difference is how accurate the distribution over parameters induced by the variational distribution is. If the distribution under the linearized mapping under the variational distribution over parameters is not accurate, this could negatively affect optimization.
>
> We compared the accuracy of the approximation of the variational distribution over functions in Appendix E.1. and E.2. and found that it is in fact highly accurate, since even with high-variance prior distributions over functions, the variance parameters of the variational distribution remain small throughout training. Please see Appendix E. for further details.
>
> We will include a more detailed discussion of this matter in the revised paper.
>
> ### 3. Comparison to Sun et al. (2019)
>
> We performed a comparison to Sun et al. and found that our method outperforms the approximation proposed by Sun et al. on 5 out 6 datasets considered. Please see our comment to all reviewers for our results.
>
> ### 4. Typos
>
> Thank you for pointing these out! We agree with all of your comments.

---

> > ### Comment · Reviewer_UY8r · 2022-08-05
> > **Thank you for your answers**
> >
> > Thank you very much for your very detailed answers to my questions and for providing additional experiments, in particular comparing your approach to that of Sun et al. (2019). Not only did you fully address my questions and concerns, I also learnt a lot. Thank you!
> >
> > Based on your replies to my and the other reviewers' questions, I keep my score and argue for acceptance.

---

> > > ### Author Response · Authors · 2022-08-05
> > > **Thank you for your feedback!**
> > >
> > > Thank you for your kind response and for recommending our manuscript for acceptance.
> > >
> > > Please don't hesitate to request further clarifications should any additional questions arise.

---

> ### Author Response · Authors · 2022-08-02
> **Response to Reviewer UY8r (1/2)**
>
> ### General Comments
>
> Thank you for your careful and positive review and for the useful pointers and suggestions.
>
> We were very excited to read that you described our paper as “an impressive paper; incredibly well-written and thorough,” containing an “impressive amount of details for making the work reproducible”.
>
> We were particularly happy that you found our work to be “novel and [a] significant contribution to the field, which will undoubtedly be of great interest to the NeurIPS community”.
>
> We hope that our response clarifies any remaining questions.
>
> ### 1. Prior variance
>
> **Quote from review:**
> > You mention that you choose a prior variance of 10, and based on the experiment in section 5.4 it is clear how this choice is seemingly the best. I am a little confused about whether this is the prior over weights of the BNNs or over functions they represent, however. I suppose it is over the weights, but the text keeps referring to the induced prior over functions. Which prior variance does Σ0 refer to?
>
> This is a good observation. In fact, for neural networks with bias parameters in the final layer and a prior with zero mean, the induced prior distribution over functions under the linearization evaluated at any finite set of points will have a zero mean and a diagonal covariance with the diagonal entries equal to the parameter variance.
>
> This is because the Jacobian of such a network evaluated at the prior mean (i.e., at zero), will be zero for all network parameters except the final-layer bias parameters, where the entries of the Jacobian will be unity. Therefore, when computing $J \Sigma J^\top$, we will get an n-by-n (for n evaluation points) covariance matrix, which, for output dimension $j$ is zero everywhere except on the diagonal, where (since $1 \times \Sigma_{j} \times 1$), the matrix will be equal to $\Sigma_{j}$. On a related note, we want to emphasize that only the distribution over **linearized** functions will have a diagonal covariance, but the distribution over non-linearized functions will not.
>
> This is non-obvious and we should have included a discussion of this somewhat odd behavior of the covariance under the linearized mapping for zero-mean priors in the paper. We have uploaded a revised version of the appendix that derives the covariance matrix in this setting in full detail.
>
> **Quote from review:**
> > Further to the question above, if Σ0 is the variance of the prior over the weights, did you try to estimate the induced variance over functions as well? Given that the datasets are normalised, an induced function-space variance of more than 1 would be peculiar.
>
> Regarding the first point, we explored prior distributions over parameters with non-zero means, so that we can obtain an induced distribution over (linearized) functions with a non-diagonal covariance function. However, except for the continual learning task, we have not been able to outperform the zero-mean prior over the network parameters in our experiments. In particular, we found that induced prior distributions over functions (induced by a prior distribution over parameters with non-zero mean) that were most similar to the diagonal distributions worked best. For example, in our experiments, we found little difference in terms of predictive performance between distributions over functions induced by a distribution over parameters with very small, non-zero mean and a variance that would induce a diagonally-dominated covariance function and a distribution over functions with a diagonal covariance whose variance is in the same order of magnitude as the diagonal of the induced non-diagonal induced covariance function.
>
> On a more general level, we want to emphasize again that it is only the distribution over **linearized** functions that will exhibit a diagonal covariance. The distribution under the corresponding non-linearized mapping will in contrast have a full covariance.
>
> Regarding the second point, we note that variance levels of priors (over functions) greater than $\approx 1$ do make sense even when the training data is normalized, since out-of-distribution data may be larger than the training data. Furthermore, for classification settings, the prior distribution over functions is defined in the logit space (i.e., before applying a softmax link function) and any finite level of prior variance in logit space is valid: A small prior variance will induce a predictive distribution with high expected entropy, whereas a large prior variance will induce a predictive distribution with high predictive variance.

---

### Author Response · Authors · 2022-08-03
**General Comments & Direct Comparison to Sun et al. (2019)**

### General Comments

We are grateful to the reviewers for their helpful and constructive feedback!

We will first make a general comment on a comparison requested by several reviewers and then refer to individual responses where we address specific comments and questions.

We greatly appreciated your questions and comments, which will greatly improve the updated paper.


### UCI Regression Comparison to Sun et al. (2019)

As requested by reviewers UY8r and UrsU, we provide an empirical comparison between our proposed method and the function-space variational inference method proposed by Sun et al. (2019).

As stated in the paper, the function-space KL approximation method proposed by Sun et al. does not scale to high-dimensional input and output data or to neural network architectures such as moderately-sized CNNs. As such, the method proposed by Sun et al. cannot be applied to image classification problems with modern neural networks. However, Sun et al. (2019) demonstrate that their proposed method performs well on a range of regression problems.

Below, we present our empirical comparison. Lower RMSE and higher log-likelihood are better. Best results are printed in bold. The first five rows are small-scale UCI experiments, and the sixth row (“Protein”) is a larger-scale experiment ($n=45,740$ data points).

**Training and evaluation protocol:** We followed the same training protocol as Sun et al. (2019) and used their code to load and process the data. The same network architecture was used (one hidden layer with 50 hidden units). We report the results for the best set of hyperparameters, computed over ten random seeds.

**Our method attains better predictive accuracy than the method proposed by Sun et al. (2019) on 5 out of 6 datasets. The difference is statistically significant (standard error shown next to mean below).**

|          |       RMSE      |       RMSE      |  Log-Likelihood  |  Log-Likelihood  |
|----------|:---------------:|:---------------:|:----------------:|:----------------:|
|          |    Sun et al.   |       Ours      |    Sun et al.    |       Ours       |
| Boston   | **2.378±0.104** | 3.632±0.515     | **-2.301±0.038** | -3.150±0.495     |
| Concrete | 4.935±0.180     | **4.177±0.443** | -3.096±0.016     | **-2.855±0.116** |
| Energy   | 0.412±0.017     | **0.409±0.060** | -0.684±0.020     | **-0.539±0.138** |
| Wine     | 0.673±0.014     | **0.615±0.033** | -1.040±0.013     | **-0.959±0.034** |
| Yacht    | 0.607±0.068     | **0.514±0.242** | -1.033±0.033     | **-0.888±0.334** |
| Protein  | 4.326±0.019     | **4.248±0.043** | -2.892±0.004     | **-2.866±0.009** |


To provide a more comprehensive picture of how our method compares to the method proposed by Sun et al. (2019) and to highlight the effect of different hyperparameters on predictive performance, **we provide the complete set of experimental results for the full range of hyperparameters tested in an interactive dashboard below**:

1. [fully-anonymized link to UCI regression on Boston Housing dataset](https://rebrand.ly/uci_boston).
2. [fully-anonymized link to UCI regression on Concrete dataset](https://rebrand.ly/uci_concrete).
3. [fully-anonymized link to UCI regression on Energy dataset](https://rebrand.ly/uci_energy).
4. [fully-anonymized link to UCI regression on Wine dataset](https://rebrand.ly/uci_wine).
5. [fully-anonymized link to UCI regression on Yacht dataset](https://rebrand.ly/uci_yacht).
6. [fully-anonymized link to UCI regression on Protein dataset](https://rebrand.ly/uci_protein).

The links above as well as the links in the individual responses below lead to interactive dashboards that show the performance of our FSVI approximation over the course of training for different hyperparameter settings. To the best of my knowledge, **the links and the dashboards are fully anonymized for either party**.

---

### Meta-Review · Area_Chair_J3qu · 2022-08-20

**Recommendation:** Accept
**Confidence:** Less certain

**Metareview:**

The review process for this manuscript is complex. The reviewers are not in consensus. Most of them have engaged considerably with the original submission as well as the significant updates that the authors have made to the manuscript post submission. In my opinion, new full covariance rank results are what make the paper interesting and these were presented after the original submission. Normally, I would find this not to be fair as the reviewers are not obligated to read such a big revision to a submitted article. But at least two reviewers have engaged with the revision considerably and I feel like the paper is stronger than what the current scores imply. The last holdout reviewer maintains a few outstanding low-confidence concerns about the paper—I do not think these should hold back the manuscript from being presented and discussed at the conference.

I am voting to accept this paper in spite of its low score, but recommend that the authors correct their behavior. Such a large revision to a manuscript puts an enormous tax on the review process; this is basically a "journal level" edit to the submission and normally this would require a second round of review.

**Award:**

No

---

### Decision · Program_Chairs · 2022-09-14

Accept